# *Drosophila* hedgehog can act as a morphogen in the absence of regulated Ci processing

**Jamie C Little†, Elisa Garcia-Garcia‡, Amanda Sul§, Daniel Kalderon***

Department of Biological Sciences, Columbia University, New York, United States

**Abstract** Extracellular Hedgehog (Hh) proteins induce transcriptional changes in target cells by inhibiting the proteolytic processing of full-length *Drosophila* Ci or mammalian Gli proteins to nuclear transcriptional repressors and by activating the full-length Ci or Gli proteins. We used Ci variants expressed at physiological levels to investigate the contributions of these mechanisms to dose-dependent Hh signaling in *Drosophila* wing imaginal discs. Ci variants that cannot be processed supported a normal pattern of graded target gene activation and the development of adults with normal wing morphology, when supplemented by constitutive Ci repressor, showing that Hh can signal normally in the absence of regulated processing. The processing-resistant Ci variants were also significantly activated in the absence of Hh by elimination of Cos2, likely acting through binding the CORD domain of Ci, or PKA, revealing separate inhibitory roles of these two components in addition to their well-established roles in promoting Ci processing.

**\*For correspondence:**
ddk1@columbia.edu

**Present address:** †Institute of Molecular Life Sciences, University of Zürich, Zürich, Switzerland; ‡Max Planck Institute for Plant Breeding Research, Cologne, Germany; §Department of Chemistry, The Scripps Research Institute, La Jolla, United States

**Competing interests:** The authors declare that no competing interests exist.

## Introduction

Hedgehog (Hh) signaling proteins guide development and help maintain adult tissue homeostasis in both invertebrates and vertebrates (*Hui and Angers, 2011*; *Ingham and McMahon, 2001*; *Petrova and Joyner, 2014*). Aberrant Hh protein production, distribution, and responses are common causes of developmental birth defects and cancer, including holoprosencephaly, limb and digit abnormalities, medulloblastoma, and basal cell carcinoma (*Anderson et al., 2012*; *Cortes et al., 2019*; *Ng and Curran, 2011*; *Pak and Segal, 2016*; *Petrova and Joyner, 2014*; *Sasai et al., 2019*). Understanding the basic molecular mechanisms of Hh communication is the first step in combating these various Hh-related disorders. Many conserved Hh components were initially identified in *Drosophila melanogaster* and then found to have a mammalian ortholog, including the key transducing protein Smoothened (Smo), which is now the target of several anticancer drugs (*Cortes et al., 2019*; *Ng and Curran, 2011*; *Pak and Segal, 2016*). There are also differences between *Drosophila* and mammalian Hh signal transduction but neither pathway is fully understood (*Briscoe and Thérond, 2013*; *Huangfu and Anderson, 2006*; *Kong et al., 2019*; *Lee et al., 2016*; *Liu, 2019*). It is therefore important to understand the fundamental molecular mechanisms involved in the pathway in *Drosophila* which is well suited to precise and detailed genetic tests conducted under physiological conditions. Hh signaling depends on a complex set of protein interactions, so it is imperative to investigate mechanisms under conditions of normal stoichiometry of signaling proteins in their normal setting.

In flies, Hh alters the interactions among a set of core signaling components to elicit the transcriptional induction and de-repression of Hh target genes through Cubitus Interruptus (Ci), the singular transcription factor of the pathway (*Domínguez et al., 1996*; *Méthot and Basler, 2001*; *Xiong et al., 2015*). Notably, Hh can act as a morphogen that signals through Ci to transcribe different Hh target gene products depending on how much ligand is present at the cell membrane. In third instar larval *Drosophila* wing discs, Hh is expressed in posterior compartment cells and Ci is

**eLife digest** Morphogens play a crucial role in determining how cells are organized in developing organisms. These chemical signals act over a wide area, and the amount of signal each cell receives typically initiates a sequence of events that spatially pattern the multiple cells of an organ or tissue. One of the most well-studied groups of morphogens are the hedgehog proteins, which are involved in the development of many animals, ranging from flies to humans.

In fruit flies, hedgehog proteins kickstart a cascade of molecular changes that switch on a set of 'target' genes. They do this by ultimately altering the activity of a protein called cubitus interruptus, which comes in two lengths: a long version called Ci-155 and a short version called Ci-75. When hedgehog is absent, Ci-155 is kept in an inactive state in the cytoplasm, where it is slowly converted into its shorter form, Ci-75: this repressor protein is then able to access the nucleus, where it switches 'off' the target genes. However, when a hedgehog signal is present, the processing of Ci into its shorter form is inhibited. Instead, Ci-155 becomes activated by a separate mechanism that allows the long form protein to enter the nucleus and switch 'on' the target genes. But it was unclear whether hedgehog requires both of these mechanisms in order to act as a morphogen and regulate the activity of developmental genes.

To answer this question, Little et al. mutated the gene for Ci in the embryo of fruit flies, so that the Ci-155 protein could no longer be processed into Ci-75. Examining the developing wings of these flies revealed that the genes targeted by hedgehog are still activated in the correct pattern. In some parts of the wing, Ci-75 is required to switch off specific sets of genes. But when Little et al. blocked these genes, by adding a gene that constantly produces the Ci repressor in the presence or absence of hedgehog, the adult flies still developed normally structured wings. This suggests that hedgehog does not need to regulate the processing of Ci-155 into Ci-75 in order to perform its developmental role.

Previous work showed that when one of the major mechanisms used by hedgehog to activate Ci-155 is blocked, fruit flies are still able to develop normal wings. Taken together with the findings of Little et al., this suggests that the two mechanisms induced by hedgehog can compensate for each other, and independently regulate the development of the fruit fly wing. These mechanisms, which are also found in humans, have been linked to birth defects and several common types of cancer, and understanding how they work could help the development of new treatments.

expressed only in anterior cells, so that Hh signals to a band of anterior cells at the anterior-posterior (AP) border with declining strength from posterior to anterior (*Blair, 2003*; *Lawrence and Struhl, 1996*). Within this AP border territory, Ci induces *decapentaplegic* (*dpp*) in a broad region, *patched* (*ptc* or a *ptc-lacZ* transcriptional reporter) in a gradient within a narrower domain, and *engrailed* (*en*) only in the cells closest to the source of Hh (see Figure 8; *Blair, 2003*; *Vervoort, 2000*). Hh controls Ci activity by regulating the processing, activation, and degradation of full-length Ci (known as Ci-155).

In the absence of Hh, the ligand-free receptor, Patched (Ptc), actively inhibits the actions of another transmembrane protein Smoothened (Smo), which is present under these conditions at relatively low levels and mainly associated with internal vesicles (*Denef et al., 2000*; *Nakano et al., 2004*; *Strigini and Cohen, 1997*; *Zhao et al., 2007*). Costal2 (Cos2), a kinesin-family protein, complexed to Fused (Fu), acts as a scaffold to bring Protein Kinase A (PKA), Glycogen Synthase Kinase-3 (GSK3), and Casein Kinase-1 (CK1) to C-155 and facilitate phosphorylation of Ci-155 at a series of clustered PKA, CK1 and GSK3 sites (*Ranieri et al., 2014*; *Zhang et al., 2005*). This creates a binding site for Slimb, the substrate recognition component of a Cul1-SCF ubiquitin ligase complex, which promotes Ci-155 ubiquitination and subsequent partial proteolysis ('processing') by the proteasome to a repressor form (Ci-75). Ci-75 lacks the C-terminal half of Ci-155, which includes its transcriptional activation domain and an epitope for a monoclonal antibody (2A1) commonly used to detect full-length Ci-155 (*Aza-Blanc et al., 1997*; *Jia et al., 2005*; *Jiang, 2006*; *Smelkinson and Kalderon, 2006*; *Smelkinson et al., 2007*). Ci-75 has a critical role in anterior wing disc cells, silencing transcription of *dpp* and *hh* (*Domínguez et al., 1996*).

Hh binding to Ptc leads to Smo activation in a process that involves Smo phosphorylation by PKA, CK1, and G-protein-coupled receptor kinase 2 (Gprk2), Smo accumulation at the plasma membrane and a change in Smo conformation or oligomerization (*Kalderon, 2008*; *Maier et al., 2014*; *Zhao et al., 2007*). Activation enhances and likely alters the nature of binding of Smo to Cos2-Fu complexes, with two important consequences. First, Ci-155 processing is inhibited, due to titration of Cos2 complexes away from Ci-155 and perhaps also to partial dissociation of PKA, CK1 or GSK3 from Cos2/Fu complexes (*Li et al., 2014*; *Ranieri et al., 2014*; *Zhang et al., 2005*). Second, Cos2-associated Fu molecules are brought together to cross-phosphorylate activation loop residues, leading to full activation of Fu protein kinase activity (*Shi et al., 2011*; *Zhang et al., 2011*; *Zhou and Kalderon, 2011*). Activated Fu protein kinase is critical for the full activation of Ci-155. If Ci-155 processing is blocked but there is no Fu kinase activity, Ci-155 is largely maintained in an inactive cytoplasmic form through direct associations with Suppressor of fused (Su(fu)) and Cos2 (*Forbes et al., 1993*; *Ohlmeyer and Kalderon, 1998*; *Préat et al., 1993*). Fu protein associations, but not kinase activity, are required for Ci-155 processing; the role of Fu kinase activity in Ci-155 activation is therefore generally studied in isolation by using point mutations in the kinase domain that only eliminate protein kinase activity and reduce Ci-155 activation (*Thérond et al., 1996*; *Zadorozny et al., 2015*).

Dose-dependent inhibition of Ci-155 processing at the AP border of wing discs might be expected to lead to a profile of increasing Ci-155 levels from anterior to posterior, with maximal levels immediately adjacent to the posterior compartment. However, Ci-155 levels actually peak near the middle of the AP border region and decline substantially over the posterior half where Hh target gene activation is strongest (*Ohlmeyer and Kalderon, 1998*; *Strigini and Cohen, 1997*). This decline is dependent on high pathway activity and is absent, for example, in wing discs lacking Fu kinase activity (*Ohlmeyer and Kalderon, 1998*). The decline in Ci-155 levels has generally been attributed to the transcriptional induction of Roadkill (Rdx), also known as Hedgehog-induced BTB protein (Hib), the substrate recognition component of a Cul3 ubiquitin ligase, culminating in the complete proteolytic destruction of ubiquitinylated Ci-155 (*Jiang, 2006*; *Kent et al., 2006*; *Zhang et al., 2006*). Loss of Rdx/Hib was initially reported to increase Ci-155 levels at the AP border (*Kent et al., 2006*; *Zhang et al., 2006*) and Rdx/Hib can target Ci-155 directly (*Zhang et al., 2009*). However, later studies reported that Rdx/Hib can also affect Ci-155 indirectly by modulating Su(fu) levels (*Liu et al., 2014*) and provided evidence that Ci-155 levels in the posterior half of the AP border of wing discs remained low in null Rdx/Hib mutant clones (*Seong et al., 2010*). Moreover, the simple idea that Rdx/Hib-induced Ci-155 degradation serves to limit pathway activity in wing discs has only limited and mixed support (*Kent et al., 2006*; *Seong et al., 2010*; *Seong and Ishii, 2013*; *Zhang et al., 2006*). Thus, the mechanisms and consequences of Hh-promoted Ci-155 reduction at the AP border remain uncertain. One indisputable consequence is that Hh-stimulated Ci-155 reduction obscures direct visualization of the pattern of Hh-inhibited Ci-155 processing at the AP border.

Ci-155 activator and Ci-75 repressor share the same zinc finger DNA-binding domain and have opposing transcriptional effects, so the concentration of each species is potentially important for all Hh target genes. However, individual target genes have different sensitivities to Ci repressor and activator depending on the arrangement of Ci binding sites and the tonic influence of other transcription factors (*Biehs et al., 2010*; *Méthot and Basler, 2001*; *Müller and Basler, 2000*; *Parker et al., 2011*). For example, repression by Ci is essential to silence *dpp* but not *ptc* or *en* in anterior cells away from the wing disc AP border. It is not clear what exactly are the spatial profiles of Ci-155 processing, Ci-155 activation or Hh-stimulated Ci-155 reduction, to what extent each regulated mechanism contributes independently to Hh morphogen action, or whether these Hh-stimulated changes are inter-dependent. To address these issues, we set out to study how processing-resistant Ci variants affected Ci-155 protein levels and activity.

The processing of Ci variants in wing discs has been investigated with some success using convenient conditions of non-physiological levels of *GAL4*-responsive *UAS*-driven transgene expression (*Jia et al., 2005*; *Smelkinson et al., 2007*). However, we previously found that Ci-155 activation, in contrast to Ci-155 processing, cannot be studied reliably in this way (*Garcia-Garcia et al., 2017*). Specifically, *ci*-null animals are very rarely rescued to adulthood using different combinations of *ci-Gal4* and *UAS-Ci* transgenes at a variety of temperatures, and anterior En expression at the AP border was not rescued in *ci*-null clones by *UAS-Ci* expressed with the commonly used wing disc driver *C765-Gal4*, with transgene expression alone sometimes eliciting a dominant-negative effect on Hh

target gene expression (*Garcia-Garcia et al., 2017*). We therefore developed genomic *ci* transgenes (*Garcia-Garcia et al., 2017*) and an efficient CRISPR strategy to directly alter the *ci* gene itself in order to study the full range of variant Ci activities under strictly physiological conditions.

In this study, we examined several processing-resistant Ci variants and found that Ci-155 protein was elevated to uniformly high levels in anterior wing disc cells away from the AP border, confirming inhibition of Ci-155 processing. At the AP border there was a prominent graded decline of Ci-155 protein from anterior to posterior for those variants fully activated by Hh, providing the clearest image yet of Hh-stimulated effects on Ci-155 levels independent of processing. Remarkably, the pattern and strength of *ptc-lacZ* and En induction in those wing discs was normal. Moreover, processing-resistant Ci variants were also found to support the development of adults with normal wing patterning, provided a constitutive source of Ci repressor was present to suppress ectopic *dpp* expression in anterior cells. Thus, Ci can mediate normal Hh morphogen action in wing discs in the complete absence of regulated processing. We also used processing-resistant Ci variants to study the effects of inhibition of processing on Ci-155 activation by Fu, as well as the potential roles of PKA and Cos2 in regulating Ci-155 activity in isolation from their well-established role in Ci-155 processing. We found that, in the absence of Hh, PKA inhibits Ci activity independent of the phosphorylation sites that regulate processing and that Cos2 inhibits Ci-155 activity, most likely by binding to the CORD region on Ci-155.

## Results

### Functional transgenes and CRISPR ci alleles

We developed strategies to study Ci expressed at physiological levels from 'genomic *ci*' transgenes (*gCi*) and CRISPR-engineered *ci* alleles (*crCi*). The former strategy used a 16 kb genomic region of *ci* that included upstream and downstream regulatory regions (*Figure 1A*) previously used within a second chromosome P-element insertion to rescue *ci* null animals (*Méthot and Basler, 1999*), inserted into an *att* site on the third chromosome (*Garcia-Garcia et al., 2017*). A *gCi-WT* transgene was readily able to rescue homozygous *ci* null (*ci⁹⁴*) animals to adulthood, with normal morphology, and behaved almost like a normal *ci* allele but with marginally lower *ci* expression and activity in wing discs (*Figure 1C,G*). We created *gCi* variants using this strategy.

We also created mutant *ci* alleles by using CRISPR in two rounds: in the first round, we put a *mini-white* marker gene in the first intron of *ci* (*Figure 1A'*); in the second round, we selected against the *mini-white* gene and introduced our mutation of interest, replacing the DNA between the first intron and the 3'UTR by homologous recombination (*Figure 1A"*). A single copy of *crCi-WT* in combination with *ci⁹⁴* resulted in efficient development of normal adults and larval wing discs with normal patterns of En, *ptc-lacZ* and Ci-155 expression in the anterior compartment (*Figure 1D,H*).

During these studies, we also became aware of an artifact, whereby low levels of ptc-*lacZ* product were detected sporadically in posterior cells of wing discs when there was a single *ci⁹⁴* allele; this occurred in flies with a normal *ci* allele (on the *Dp(y⁺)* 'balancer') (*Figure 1E*) or with the *crCi-WT* allele (*Figure 1D*). The artifact was also seen with *gCi-WT* when *ci⁹⁴* was heterozygous (data not shown) but not when *ci⁹⁴* was homozygous (*Figure 1C*) or when *crCi-WT* was homozygous (*Figure 1B*). We sequenced the relevant region of the *ci⁹⁴* allele and confirmed that it was the same deletion originally reported (*Méthot and Basler, 1999*; *Slusarski et al., 1995*) and in FlyBase. We also induced homozygous *ci⁹⁴* clones (by *FRT*-mediated recombination to remove a second chromosome genomic *ci* transgene) and confirmed that *ci⁹⁴* encoded no detectable Ci-155 protein (data not shown). Thus, despite the observed sporadic expression of *ptc-lacZ* in posterior cells in some genetic backgrounds, we are confident that the activity of Ci variants can be assayed in anterior wing disc cells in a null background under physiological conditions using either *crCi* alleles or *gCi* transgenes.

### Processing-resistant Ci variants have elevated Ci-155 levels in anterior cells

PKA phosphorylates Ci-155 at amino acids S838, S856, and S892 to create recognition sites for both GSK3 and CK1, which further phosphorylate Ci-155 at a consecutive series of primed phosphorylation sites (*Smelkinson and Kalderon, 2006*; *Smelkinson et al., 2007*). The phosphorylation series

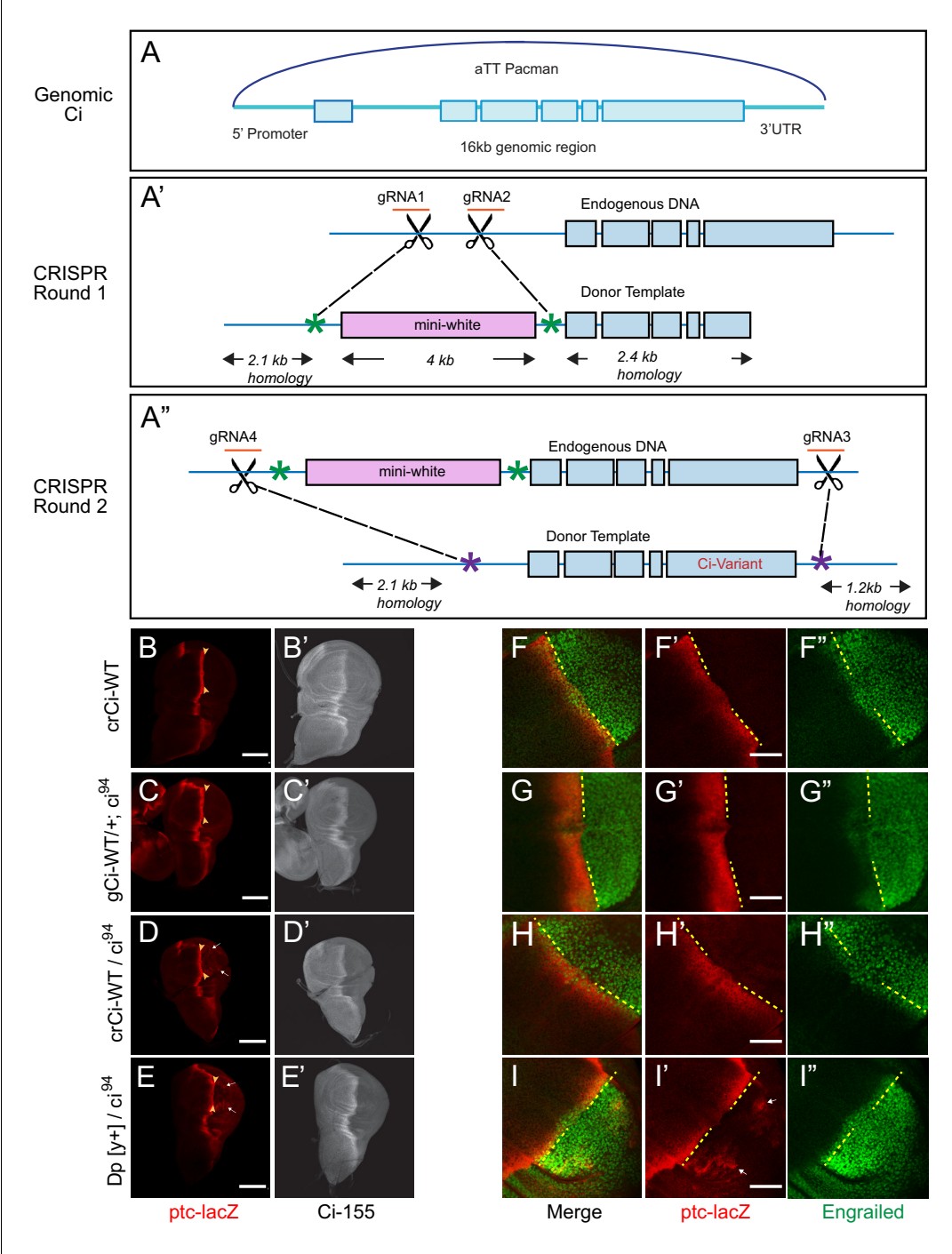

**Figure 1.** Wild-type genomic ci transgene and CRISPR ci alleles are fully functional. (**A**) The 'genomic Ci' transgene (*gCi*) was derived from a 16 kb genomic region of *ci* cloned into an att-Pacman vector and inserted at *att ZH-86FB* located at 86F on the third chromosome. It includes a 6.7 kb upstream promoter region, exons1-6 (blue boxes), introns (blue line), and 0.7 kb of the 3'UTR. (**A'–A"**) CRISPR *ci* (*crCi*) alleles were generated in two rounds. (**A'**) The first round inserted a mini-white gene (pink box) into the first intron of endogenous *ci* using two guide RNAs (orange) in the first intron and altered PAM sites on the donor template (green star). (**A"**) The second round replaced intron 1 and exons 2–6 (blue lines and boxes) including the mini-white gene; the donor template had mutated PAM sites (purple stars) corresponding to the gRNA4 site, approximately 30 bp outside the mutated PAM site for gRNA1, and in the 3'UTR 5 kb away from gRNA 2, labeled gRNA 3. (**B–E**) Third instar wing discs showing *ptc-lacZ* reporter gene expression, visualized by Beta-galactosidase antibody staining (red), with the posterior edge of AP border expression marked by yellow arrowheads, and (**B'–E'**) full-length Ci-155, visualized by 2A1 antibody staining (gray-scale). Anterior is left and ventral is up. (**B**) Two copies and (**D**) one copy of *crCi-WT*, or (**C**) one copy of *gCi-WT* supported normal patterns of elevated *ptc-lacZ* and Ci-155 at the AP border but (**D, E, I'**) sporadic ectopic posterior

*Figure 1 continued on next page*

Figure 1 continued

*ptc-lacZ* expression (white arrows) was seen whenever a single *ci94* allele was present, even (E) in discs with no synthetic *ci* transgene or allele (*Dp[y+]* has wild-type *ci*). (F–I) Induction of En (green) at the AP border was detected by using the posterior boundary (yellow dashed line) of *ptc-lacZ* (red) to distinguish anterior (left) from posterior compartment cells, which express En independent of Hh signaling. En induction was normal in the presence of (F) two copies of *cr-Ci-WT*, (H) one copy of *crCi-WT* or (I) one wild-type *ci* allele and (G) was slightly reduced in the presence of one copy of *gCi-WT*. Scale bars are (B–E) 100 μm and (F–I) 40 μm.

creates a binding site for Slimb that includes the core peptide pSpTYYGpS$_{849}$MQpS, spanning residues 844–852. Ci-S849A lacks the last CK1 target site initially primed by PKA phosphorylation of S838 and Ci-P(1-3)A has alterations to all three PKA sites (S838A, S856A, and S892A). Ci fragments with those alterations showed complete loss of Slimb binding in vitro after phosphorylation by PKA, CK1 and GSK3, while *UAS-Ci* transgene products with those changes showed no processing in wing discs, judged by a sensitive assay of repressor function in posterior compartment wing disc cells (*Smelkinson and Kalderon, 2006*; *Smelkinson et al., 2007*). The activity of these proteins has not previously been measured under physiological conditions. We therefore used *ci* alleles with those alterations to determine how loss of processing affects Ci protein levels and activity at normal physiological levels.

The wing discs of animals with *crCi-S849A* or *crCi-P(1-3)A* in combination with *ci94* had expanded anterior regions (*Figure 2A–D*), as expected because Ci-75 repressor, normally produced from Ci-155 processing, is required to silence *dpp* expression in anterior cells and ectopic anterior Dpp induces anterior growth (*Méthot and Basler, 1999*). We also found that these wing discs had strongly elevated Ci-155 levels throughout the anterior, indicating that full-length Ci-155 was not being processed in the absence of the Hh signal, as expected (*Figure 2A–D*; *Figure 2—figure supplement 1A–D,G,H*).

## Processing-resistant Ci variants reveal the pattern of Hh-stimulated Ci-155 reduction at the AP border

Although normal wing discs have a clear stripe of elevated Ci-155 at the AP border relative to anterior cells (*Figure 2A*), Ci-155 levels actually decline over the posterior half of the AP border (*Figure 2G*) in a manner that depends on strong activation of the Hh pathway (*Ohlmeyer and Kalderon, 1998*; *Strigini and Cohen, 1997*). Although Ci-155 protein levels were strongly elevated compared to normal for Ci-S849A and Ci-P(1-3)A in anterior cells, there was a sharp decline toward the posterior of AP border territory (*Figure 2H,I*; *Figure 2—figure supplement 1A–D,G,H*). This profile represents the gradient of Hh-stimulated Ci-155 loss that has generally been attributed to full degradation. It has not previously been seen in isolation because it is normally super-imposed on an unknown profile due to inhibition of Ci processing for wild-type Ci (*Figure 2K*). Moreover, if the observed pattern of Ci-155 reduction is the same for wild-type Ci (see later), we can subtract this profile from the observed Ci-155 profile of wild-type Ci to deduce a profile of wild-type Ci-155 processing (*Figure 2L*). The result shows that the inhibition of Ci-155 processing is graded, with a spatial profile broadly similar to that of *ptc-lacZ* activation, but with a slightly higher sensitivity to low levels of Hh (*Figure 2L*). Thus, comparison of the Ci-155 profiles of wild-type and processing-resistant variants provided the best evidence to date of the spatial patterns of graded inhibition by Hh of Ci-155 processing (*Figure 2L*) and of graded, Hh-promoted Ci-155 loss at the AP border (*Figure 2H,I,K*).

We also created a *ci* allele, CiΔ1270–1370, resembling a C-terminal deletion variant that had previously been found not to undergo processing in assays using cultured cells and *UAS-Ci* transgenes in wing discs (*Wang and Price, 2008*; *Zhou and Kalderon, 2010*). CiΔ1270–1370 also had uniformly elevated Ci levels in anterior wing disc cells, consistent with a lack of processing (*Figure 2E,J*; *Figure 2—figure supplement 1E*). However, unlike Ci-S849A and Ci-P(1-3)A, this Ci variant induced *ptc-lacZ* and En only weakly at the AP border (*Figure 2A–E,G–J*; *Figure 3A–C*; *Figure 2—figure supplement 1P*; *Figure 2—figure supplement 2E*). There was also no decline of Ci-155 protein within AP territory (*Figure 2J,K*), consistent with prior evidence that Hh-stimulated Ci-155 reduction, visualized clearly with the other processing-resistant Ci variants, is only observed at high levels of Hh signaling.

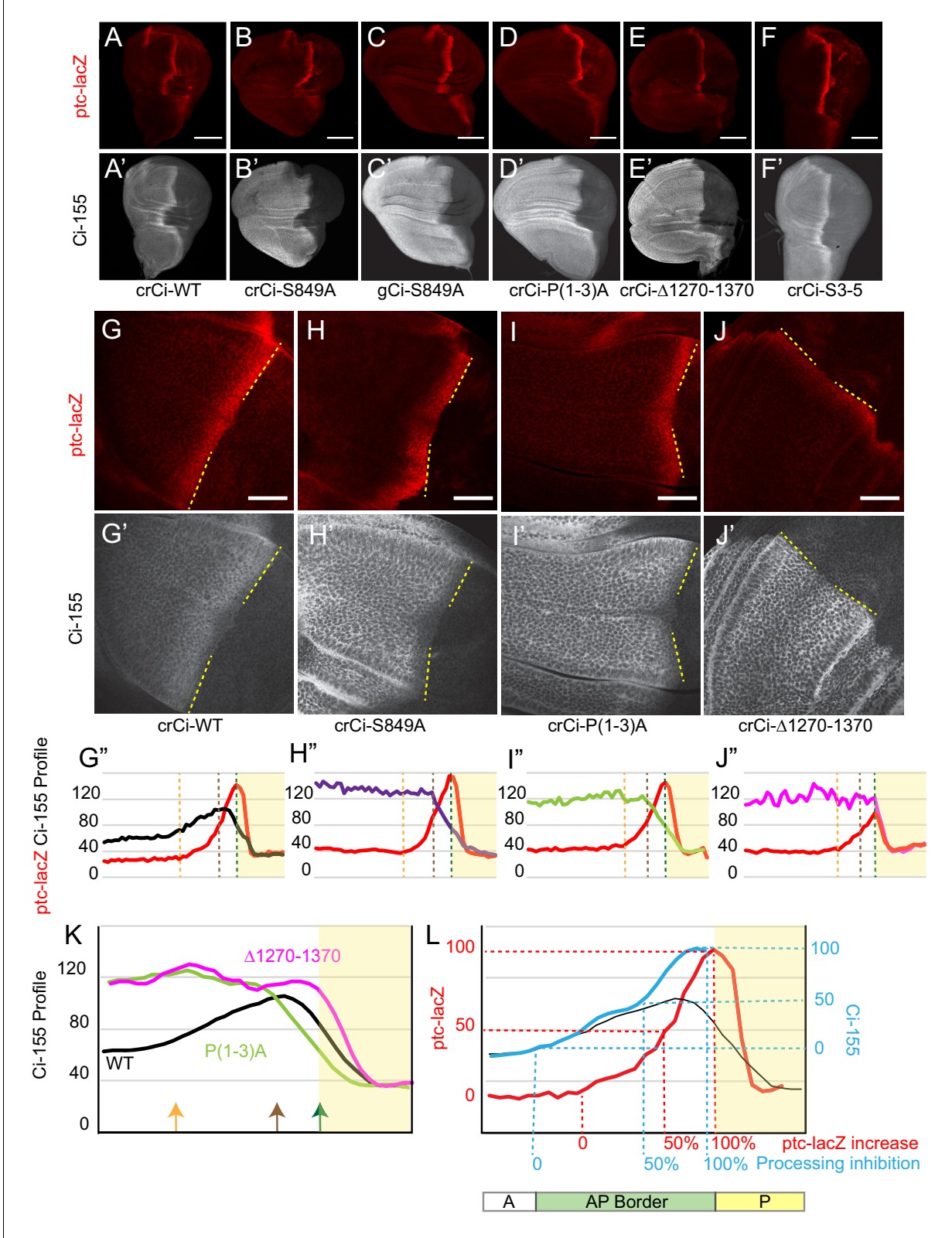

**Figure 2.** Processing-resistant Ci variants reveal the gradients of Hh-stimulated Ci-155 degradation and Hh inhibition of Ci-155 processing. (**A–J**) *ptc-lacZ* (red) and (**A'–J'**) Ci-155 (gray-scale) in wing discs with one copy of the indicated *crCi* alleles (with *ci*[94]) at (**A–F**) low (20x objective) and (**G–J**) high (63x objective) magnification, with AP boundary (dotted yellow line at posterior *ptc-lacZ* boundary). Scale bars are (**A–F**) 100 µm and (**G–J**) 40 µm. (**G''–J''**) Intensity profiles for *ptc-lacZ* (red) and Ci-155 (black) from anterior (left) to posterior. Vertical lines indicate *ptc-lacZ* peak (green), initial rise (orange)

*Figure 2 continued on next page*

Figure 2 continued

and 50% increase to peak (brown). Profiles are from two wing discs for *crCi-Δ1270–1370* and three discs for all other samples, aligned and measured as described in Materials and methods. Note that the green line corresponding to maximal *ptc-lacZ* effectively represents the AP compartment boundary. The profile of *ptc-lacZ* posterior to that location does not decline precipitously but the decline is not informative (it likely results in part because the columnar cells are not uniformly shaped, so that the measured z-sections include portions of anterior and posterior cells). Territory posterior to the *ptc-lacZ* peak has yellow shading in (G"–J") and (K, L) to indicate that it does not contain useful information. The profiles of *ptc-lacZ* and Ci-155 that report responses to Hh are in the territory anterior to the *ptc-lacZ* peak. (K) Normalized Ci-155 profiles for indicated *crCi* alleles derived from G'-J' but with a smoothing function that calculates average intensity for five successive locations centered on each x-axis location. Arrows indicated locations of *ptc-lacZ* initial rise, 50% increase and peak for *crCi-WT* discs. (L) *ptc-lacZ* (red) and Ci-155 (black) smoothened profiles for *crCi-WT*, with red guide lines for locations of initial rise, 50% increase and peak *ptc-lacZ*. The difference between the average Ci-155 intensity for Ci-P(1-3)A and Ci-S849A at each point along the x-axis was subtracted from the maximum Ci-155 intensity for those genotypes (observed in cells anterior to the AP border) to calculate values for Hh-stimulated Ci-155 reduction. These values were added to the Ci-WT Ci-155 profile at each location to produce the blue curve, representing Ci-155 levels in the absence of Hh-stimulated reduction. Blue guide lines show the locations where inferred Ci-155 processing is first inhibited, 50% inhibited, and fully inhibited. See also *Figure 2—figure supplement 1* and *Figure 2—figure supplement 2*.

The online version of this article includes the following source data and figure supplement(s) for figure 2:

**Source data 1.** Numerical data for graphs in *Figure 2*.
**Source data 2.** Numerical data for graphs In *Figure 2*.
**Figure supplement 1.** Loss of Hib binding sites does not greatly affect Ci-155 activity or Hh-stimulated proteolysis.
**Figure supplement 1—source data 1.** Numerical data for graphs in *Figure 2—figure supplement 1*.
**Figure supplement 2.** Processing-resistant Ci-155 induces ectopic dpp expression and anterior disc expansions are suppressed by adding a constitutive Ci repressor.

## Hh-promoted Ci-155 reduction is not eliminated by altering major Rdx/Hib-binding sites

To study Hh-promoted Ci-155 reduction further we created an allele encoding a Ci variant with compromised Rdx/Hib binding. Rdx/Hib binds to Ci-155 through multiple sites; altering three principal binding regions (designated S3,4,5 in the cited study) through clustered point mutations rendered the altered Ci-155 ('Ci-S3-5') largely insensitive to Rdx/Hib in a tissue culture assay (*Zhang et al., 2009*). We found that animals expressing one copy of *crCi-S3-5* (in combination with *ci$^{94}$*) developed efficiently into adults with normally patterned wings (data not shown). In larval wing discs, the peak of *ptc-lacZ* expression was slightly elevated at the AP border compared to normal but the domain of induction of En (a high-level Hh target) was not expanded (*Figure 2F*; *Figure 2—figure supplement 1F,I,M–O*). The Ci-155 profile included low anterior levels, suggesting normal processing, and declined in the posterior regions of the AP border much like wild-type Ci-155, showing that Hh-stimulated Ci-155 reduction remained robust (*Figure 2F*; *Figure 2—figure supplement 1I,L*). The properties of Ci-S3-5 suggest that direct action of Rdx/Hib on Ci-155 does not account for a significant fraction of the reduction of Ci-155 stimulated by the highest levels of Hh signaling. Previous studies have not specifically tested only the direct effects of Rdx/Hib on Ci-155 in wing discs; some studies found that elimination or reduction of Rdx/Hib activity increased Ci-155 levels (*Kent et al., 2006*; *Zhang et al., 2006*), while others found no change in Ci-155 levels in the posterior half of the AP border region (*Seong et al., 2010*; *Seong and Ishii, 2013*).

## Su(fu) is involved in Hh-promoted Ci-155 reduction at the AP border

Suppressor of fused (Su(fu)) may participate in the Hh-stimulated reduction of Ci-155 at the AP border, potentially in more than one way. It has been found that Rdx/Hib indirectly reduces Su(fu) protein levels at the AP border (*Liu et al., 2014*) and it has been suggested that Su(fu) competes with Rdx/Hib for Ci-155 binding (*Zhang et al., 2006*). It has also been shown that loss of Su(fu) leads to greatly reduced Ci-155 levels, presumed to be due to enhanced degradation of Su(fu)-free Ci-155, throughout the wing disc (*Ohlmeyer and Kalderon, 1998*) and it has been conjectured that Hh may activate Ci-155 in part through Su(fu) dissociation from Ci-155, as suggested by studies of Gli activation (*Humke et al., 2010*; *Lee et al., 2016*; *Tukachinsky et al., 2010*).

We examined Ci-155 AP border profiles for wild-type Ci and Ci-S3-5 in the absence of Su(fu). The two profiles were extremely similar; Ci-155 levels appeared to peak at, or very close to the AP compartment boundary, suggesting little or no Hh-stimulated loss (*Figure 2—figure supplement 1J–L*). The results are consistent with the hypothesis that Su(fu) is a key factor in the regulation of Hh-

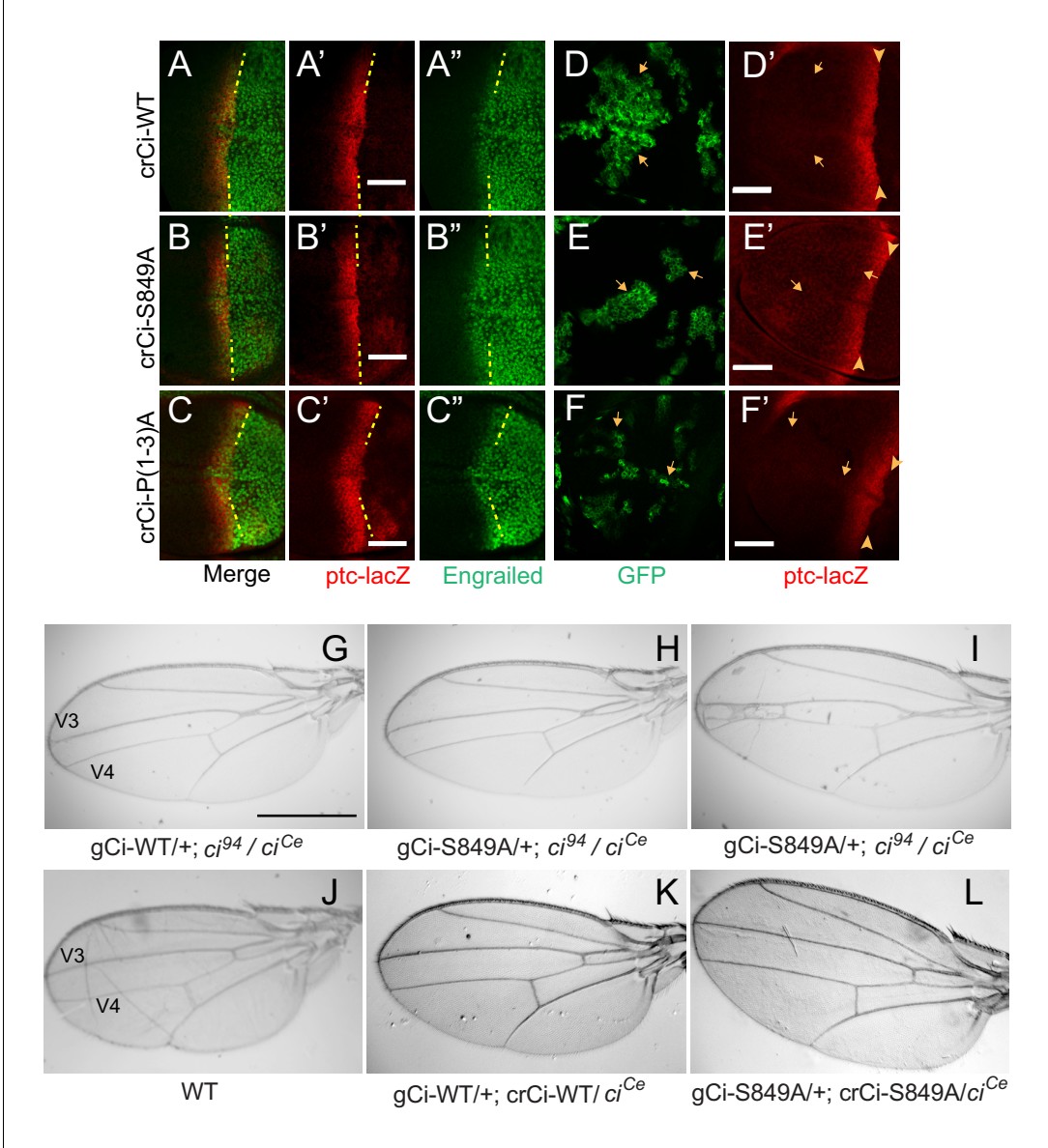

**Figure 3.** Processing-resistant Ci variants support normal Hh signaling and wing patterning (**A–C**) Wing discs with one copy of indicated *crCi* alleles. *ptc-lacZ* (red) indicates the AP compartment boundary (yellow line) to reveal induction of the high-level Hh target gene En (green) in anterior cells at the AP border. (**D–F**) Wing discs with anterior clones (GFP, green, yellow arrows) that have lost a second chromosome *gCi* transgene, leaving one copy of the indicated *crCi* alleles as a source of Ci. (**D'–F'**) Little (**E'**) or no (**D', F'**) *ptc-lacZ* induction was observed in the clones (arrows) relative to the AP border (arrowheads). Scale bars are (**A–F**) 40 μm. (**G–L**) Wings from adult flies with the indicated *ci* transgenes and alleles (*ci^{Ce}* encodes a constitutive repressor). The spacing between veins 3 and 4 is (**J–L**) normal for two copies of WT or S849A *ci* alleles and (**G–I**) similarly reduced for one copy of WT or S849A alleles. At least five high-quality mounted wings were examined for each genotype. Scale bars are (**G–I**) 500 μm.

stimulated Ci-155 reduction at the AP border. This role of Su(fu) was apparent even in the absence of normal Rdx/Hib binding to Ci-155, suggesting that Su(fu) is not acting principally by competing with Rdx/Hib for Ci-155 binding.

## Processing-resistant Ci variants have normal activity at the AP border

Remarkably, the pattern of En and *ptc-lacZ* induction at the AP border was normal for Ci-S849A and Ci-P(1-3)A (*Figure 3A–C*; *Figure 2—figure supplement 1P*), showing that Ci-155 processing is not essential for dose-dependent induction of these Hh target genes. The unchanged profile of pathway activity suggests that the profile of pathway-induced Ci-155 reduction is also likely to be the same for wild-type Ci and processing-resistant Ci variants, supporting the validity of using the latter profile to deduce the processing pattern of wild-type Ci-155 (*Figure 2L*).

Ci-S849A and Ci-P(1-3A) wing discs expressed *dpp* ectopically in anterior cells, as expected from the absence of Ci-75 repressor (*Figure 2—figure supplement 2A,B*). The expanded anterior regions of these wing discs are likely responsible for the failure to recover adults expressing only processing-resistant Ci variants, precluding analysis of adult wing patterning. The addition of a *ci^Ce* allele, which encodes a constitutive repressor form of Ci (and no activator) (*Méthot and Basler, 1999*; *Slusarski et al., 1995*), restored normal wing disc morphology without significantly affecting *ptc-lacZ* expression at the AP border (*Figure 2—figure supplement 2C,D*) and allowed recovery of adults.

Normal wing morphology depends on long-range patterning elicited by the central stripe of Hh-induced Dpp and on creation of a central inter-vein region between veins 3 and 4 by stronger Hh signaling, sufficient to induce the transcription factor Collier, also known as Knot (*Mohler et al., 2000*; *Vervoort, 2000*; *Vervoort et al., 1999*). The adult wing phenotypes of animals with one copy of gCi-WT or gCi-S849A in a *ci^94/ci^Ce* background were similar to each other, with a consistent moderate pinching between veins 3 and 4 (*Figure 3G,H*), although some animals with gCi-S849A also showed a greater narrowing of the inter-vein region (*Figure 3I*). We then tested the activity of a *gCi* transgene together with a *crCi* allele in trans to *ci^Ce*. We found that wing morphology was absolutely normal for flies with both *gCi* and *crCi* encoded wild-type Ci or when both encoded processing-resistant Ci-S849A (*Figure 3J–L*). Hence, we conclude that Hh can fulfill its normal morphogenetic function, culminating in a normally patterned wing in the complete absence of regulated Ci-155 processing.

## Dependence of Ci-155 activity induced by Fused kinase on inhibition of Ci-155 processing

Fu can be activated synthetically in the absence of Hh stimulation by overexpression of Fu variants with either a membrane-targeting tag (GAP-Fu) or acidic residue replacements of phosphorylation sites key to normal activation (Fu-EE) (*Claret et al., 2007*; *Zhou and Kalderon, 2011*). Activated Fu can partially activate Smo (*Claret et al., 2007*; *Sanial et al., 2017*) but direct downstream, Smo-independent actions can be measured by assaying responses in *smo* mutant anterior clones expressing Fu-EE or GAP-Fu. Previously, such experiments showed that activated Fu alone was sufficient to elicit strong Hh target gene induction, suggesting that Ci-155 activation can be effective even without the normal inhibition of processing that occurs at the AP border (*Zhou and Kalderon, 2011*). Fu kinase activity is not required for Hh to block Ci-155 processing at the AP border (*Alves et al., 1998*; *Ohlmeyer and Kalderon, 1998*; *Zadorozny et al., 2015*). Nevertheless, synthetically activated Fu was observed to increase Ci-155 levels in anterior clones and further tests suggested this likely resulted from partial inhibition of Ci-155 processing mediated by Cos2 phosphorylation (*Zhou and Kalderon, 2011*).

To clarify the dependence of Ci-155 activation by Fu on Ci-155 processing inhibition we compared the activities of wild-type and processing-resistant Ci variants in *smo* mutant clones expressing activated GAP-Fu. We found that Ci-S849A or CiΔ1270–1370, provided by a single *crCi* allele in trans to *ci^94*, mediated *ptc-lacZ* induction in anterior *smo GAP-Fu* clones to the same level as at the AP border, whereas induction mediated by a single wild-type *crCi* allele was much lower (about 50%) (*Figure 4A–D,J*). In each case, the activity in clones was compared to the AP border of the same wing discs and reflects the activity of the same source of Ci. Similar results were seen in GAP-Fu clones that retained a functional *smo* allele, with *ptc-lacZ* induction of the three processing-resistant variants (Ci-S849A, Ci-P(1-3)A, and CiΔ1270–1370) greatly exceeding that of wild-type Ci (*Figure 4F–I,K*). Thus, Hh target gene induction by Fu kinase alone was quite weak in the presence of a single wild-type *ci* allele and was substantially increased if Ci-155 processing was also inhibited. The observed increase could in principle be due to an increased supply of Ci-155 or loss of Ci-75

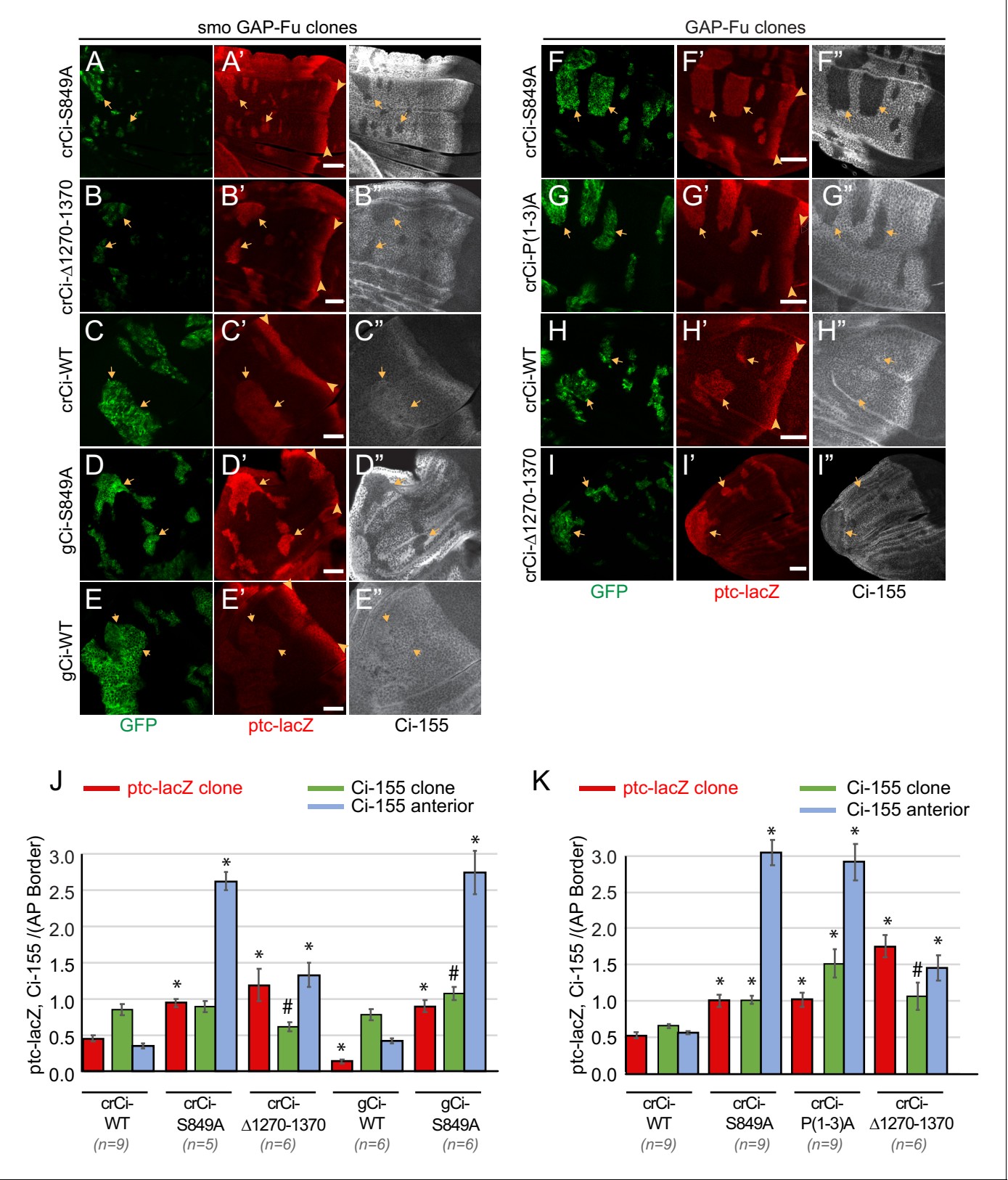

**Figure 4.** Activation by Fused kinase is enhanced by blocking Ci-155 processing. (A–I) Wing discs from animals with one copy of the designated *ci* transgenes and alleles (together with *ci⁹⁴*) with clones (GFP, green, arrows) that express *UAS-GAP-Fu* and (A–E) lack *smo* activity or (F–I) are heterozygous for *smo* (arrowheads indicate AP border), showing (A'–I') *ptc-lacZ* (red) and (A"–I") Ci-155 (gray-scale). (A"–I") Ci-155 levels were much

*Figure 4 continued on next page*

*Figure 4 continued*

reduced in clones whenever pathway activity was strongly induced (**A″**, **B″**, **D″**, **F″**, **G″**). Scale bars are 40 µm. (**J**, **K**) Average intensity of *ptc-lacZ* in clones (red), Ci-155 in clones (green) or neighboring anterior territory (blue), as a fraction of AP border levels for (**J**) *smo GAP-Fu* clones and (**K**) *GAP-Fu* clones. Mean and SEM shown. Significant differences between values for a given genotype compared to those for *crCi-WT*, calculated by paired t-tests, are indicated for $p < 0.001$ (*) and $p < 0.05$ (#). Additionally, in (**J**) *ptc-lacZ* was significantly increased for *gCi-S849A* versus *gCi-WT* ($p < 0.0001$), as was the anterior level of Ci-155 ($p < 0.0001$).

The online version of this article includes the following source data for figure 4:

**Source data 1.** Numerical data for graphs in *Figure 4*.

repressor, or both. When slightly lower levels of wild-type Ci protein were provided by a single *gCi* transgene instead of a *cr-Ci* allele, GAP-Fu induced significantly lower levels of *ptc-lacZ* (*Figure 4J*), suggesting that the supply of Ci-155 is a key factor. Thus, producing a robust supply of Ci-155 that is not substantially diminished by processing to Ci-75 is important for Fu to elicit high Ci-155 activity. This dependence was highlighted by using only a single functional *ci* allele and by assaying synthetically activated Fu in anterior cells.

The Ci-155 levels detected within *smo GAP-Fu* clones were much lower than in surrounding territory in wing discs expressing processing-resistant Ci variants (*Figure 4A,B,D,J*). The reduction in Ci-155 was similar in magnitude to that observed in posterior regions of the AP border and presumably reflects Ci-155 reduction due to high Hh pathway activity. By contrast, wild-type Ci-155 levels were elevated in *smo GAP-Fu* clones relative to neighboring cells. Since *ptc-lacZ* in these clones was significantly lower than at the AP border (*Figure 4C,J*) there is likely little or no reduction in Ci-155 due to high pathway activity. The fact that Ci-155 clone levels were lower than maximal AP border Ci-155 levels for wild-type Ci (*Figure 4C*) therefore indicates that GAP-Fu does not inhibit processing to the same degree as Hh inhibits processing at the AP border. Thus, although the absolute steady-state levels of Ci-155 for Ci-WT and Ci-S849A in *smo GAP-Fu* clones were quite similar (*Figure 4J*), Ci-155 accumulation was limited by largely different mechanisms; significant continued processing for Ci-WT and pathway-stimulated loss for Ci-S849A.

In summary, activation of Ci-155 by Fu to produce high levels of Hh target gene expression also requires provision of high levels of primary Ci-155 translation product that is protected from processing. The elevated Ci-155 supply produced by processing-resistant Ci variants is, however, not directly evident from measurement of steady-state Ci-155 levels because of subsequent, robust Ci-155 loss in response to high pathway activity. Even though steady-state Ci-155 levels are similar for Ci-WT and Ci-S849A, the proportion of Ci-155 molecules that are active is presumably higher for Ci-S849A in GAP-Fu clones.

## PKA and Cos2 silence Ci-155 activity

It was previously appreciated that Cos2, PKA and Slimb are all necessary for Ci-155 processing but that induction of Hh target genes was higher in anterior *cos2* and *pka* mutant clones than in *slimb* mutant clones (*Jiang and Struhl, 1998*; *Smelkinson et al., 2007*; *Wang et al., 1999*). Loss of PKA also increased *ptc-lacZ* induction in *slimb* mutant clones (*Smelkinson et al., 2007*). These observations suggested that PKA and Cos2 inhibit Ci-155 activity in addition to promoting Ci-155 processing, with the potential reservations that the *slimb* alleles used in some tests may not have fully blocked Ci-155 processing or that Slimb may have additional relevant actions that reduce Ci-155 activity. The effect of PKA loss on the activity of processing-resistant *UAS-Ci* transgenes has also been investigated previously but the transgenes were expressed at non-physiological levels and such transgenes do not support normal Hh responses at the AP border (*Smelkinson et al., 2007*; *Garcia-Garcia et al., 2017*).

To test the effects of PKA and Cos2 on the activity of processing-resistant Ci-155 expressed at physiological levels, we induced *pka* or *cos2* clones in wing discs expressing Ci-P(1-3A) from a single allele in combination with *ci94*. We found that in both types of clone, there was a marked increase of *ptc-lacZ* expression compared to surrounding tissue (*Figure 5B,C,I,L*) and compared to clones with no change in PKA or Cos2 activities (*Figure 3F*). The level of *ptc-lacZ* induced was found to be about 75% (*pka* clones) or 50% (*cos2* clones) of AP border levels in the same wing discs (*Figure 5L*). The levels of *ptc-lacZ* induced in equivalent clones in wing discs expressing one allele of wild-type Ci were very similar (*Figure 5A,H,L*), indicating that the activity of Ci-P(1-3A) reflected the normal

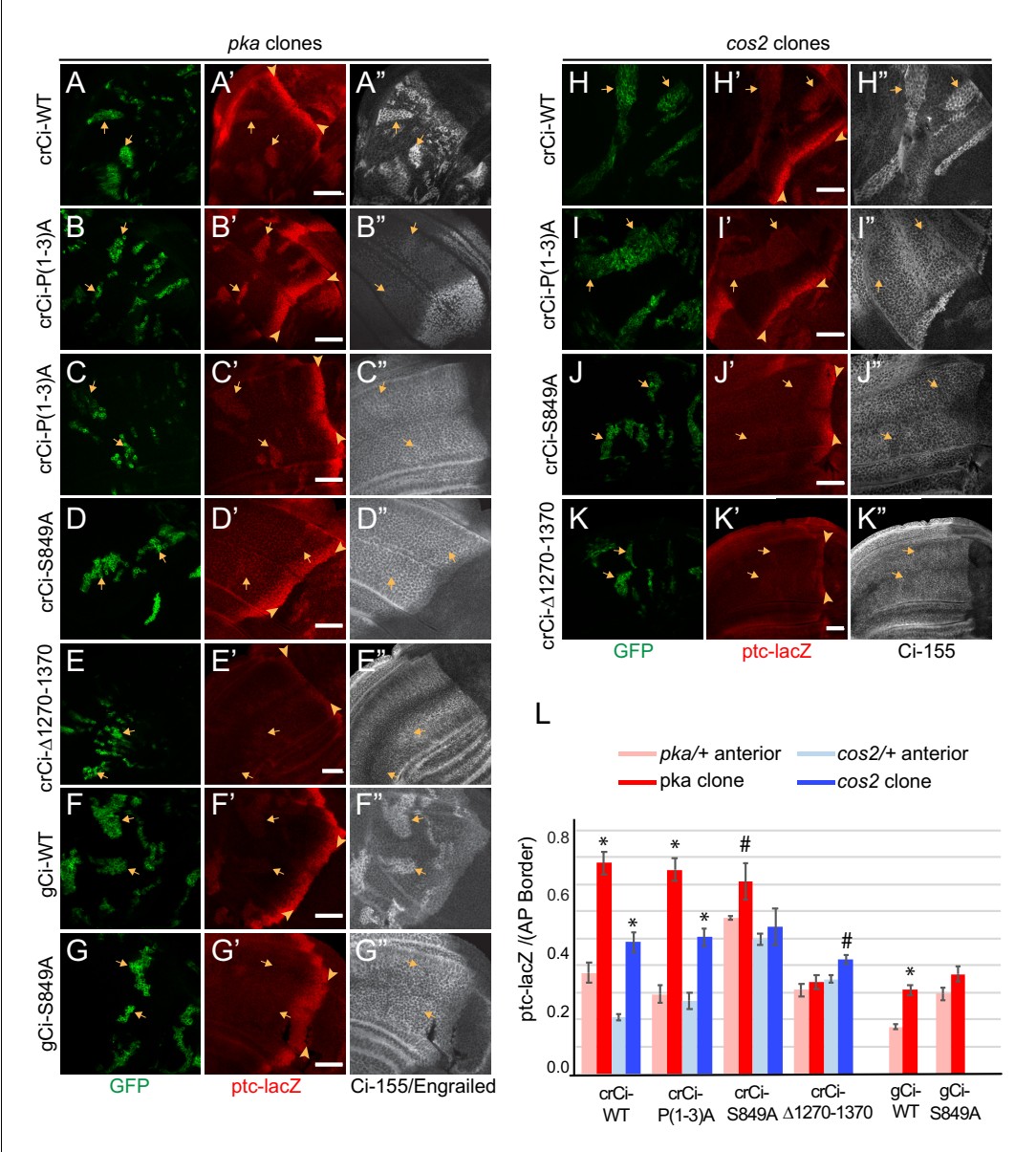

**Figure 5.** PKA and Cos2 reduce the activity of Ci-155 that is not processed. (**A–K**) Wing discs from animals with one copy of the designated *ci* transgenes and alleles (together with *ci^94*) with clones (GFP, green, arrows) that lack (**A–G**) *pka* activity or (**H–K**) *cos2* activity (arrowheads indicate AP border), showing (**A'–K'**) *ptc-lacZ* (red) and (**A", C"–K"**) Ci-155 (gray-scale). (**B"**) En (gray-scale) was weakly induced in *pka* clones from discs expressing Ci-P(1-3)A. Ci-155 levels were (**A", F", H"**) increased relative to neighboring anterior territory for Ci-WT but were (**C"–E", G", I"–K"**) either unchanged or slightly reduced, presumably from full proteolysis, for processing-resistant Ci variants. Scale bars are 40 µm. (**L**) Average intensity of *ptc-lacZ* in *pka* clones (red) or neighboring anterior territory (pink), and in *cos2* clones (dark blue) or neighboring anterior territory (light blue), as a fraction of AP border levels. Mean and SEM shown. Significant differences between *ptc-lacZ* values in *pka* or *cos2* mutant clones and neighboring anterior *pka/+* or *cos2/+* cells for a given genotype, calculated by paired t-tests, are indicated for p<0.001 (*) and p<0.05 (#).

The online version of this article includes the following source data for figure 5:

**Source data 1.** Numerical data for graphs in *Figure 5*.

response of wild-type Ci to loss of PKA or Cos2 and hence that the three PKA sites that are key for processing (P1-3) are not required for the regulation of Ci-155 activity by PKA or Cos2.

Induction of *ptc-lacZ* was substantially lower for *pka* mutant clones expressing wild-type Ci from a *gCi* transgene rather than from a *cr-Ci* allele (*Figure 5F,L*), showing that Ci-155 activity elicited by loss of PKA depends on Ci-155 levels. This dependence was previously shown by comparing wild-

type animals and *ci* heterozygotes, and it was further shown that Hh target gene induction depended on the relative stoichiometry of Ci-155 and Su(fu), suggesting the hypothesis that only Su (fu)-free Ci-155 is active in *pka* mutant clones (*Ohlmeyer and Kalderon, 1998*). In all cases (*pka* or *cos2* mutant clones, Ci-WT or Ci-P(1-3A), gCi-WT or crCi-WT), the levels of Ci-155 in clones matched or exceeded the highest levels at the AP border, suggesting little or no loss of Ci-155 due to high pathway activity. Indeed, activity in *pka* and *cos2* clones may depend on Ci-155 levels exceeding the inhibitory capacity of Su(fu). No such requirement is expected in GAP-Fu clones because Fu can relieve inhibition by Su(fu). Thus, in contrast to the situation with GAP-Fu clones, the contribution to activity of a robust supply of Ci-155 that is not processed is reflected in elevated steady-state Ci-155 levels. In summary, the results for Ci-P(1-3A) clearly indicate that PKA and Cos2 inhibit the activity of Ci-155 that is not processed in the absence of Hh stimulation. The magnitude of inhibition is substantial.

Surprisingly, *ptc-lacZ* induction by Ci-S849A was not clearly higher in *pka* or *cos2* mutant clones than in surrounding cells (*Figure 5D,G,J*). Quantitation revealed that this was largely due to significant *ptc-lacZ* expression in heterozygous tissue surrounding the clones (*Figure 5L*). Similar, low levels of *ptc-lacZ* activity were observed also for Ci-S849A, but not Ci-WT or Ci-P(1-3A), in clones with normal PKA and Cos2 activity (*Figure 3D–F*). These results suggest that S849 is relevant to the regulation of Ci-155 activity by PKA and Cos2.

CiΔ1270–1370, which is also not subject to processing, did not induce *ptc-lacZ* in wild-type anterior cells (*Figure 2E,J*; *Figure 2—figure supplement 2E*) or in *pka/+* or *cos2/+* cells and was not strongly activated by loss of PKA or Cos2, producing *ptc-lacZ* expression significantly lower than Ci-WT (*Figure 5E,K,L*). CiΔ1270–1370 also has lower activity than Ci-WT at the AP border of wild-type discs (*Figure 2E,J*), but it is strongly activated by GAP-Fu (*Figure 4B,I–K*). Based on these observations and the hypothesis that PKA and Cos2 primarily inhibit Su(fu)-free Ci-155, we speculate that CiΔ1270–1370 may be inhibited more strongly than wild-type Ci by Su(fu), so that release from PKA and Cos2 inhibition is without effect and full relief from Su(fu) inhibition is achieved only by artificially strong GAP-Fu activation and not by normal Fu activation at the AP border.

## Cos2 likely silences Ci activity by binding to the CORD region

To investigate how Cos2 silences Ci-155 activity, we further examined the interactions between these proteins. Cos2 can bind to Ci-155 through three regions defined by in vitro binding assays: the CDN region (residues 346–440), the zinc fingers (residues 506–620) and the CORD domain (residues 934–1065) (*Wang and Jiang, 2004*; *Zhou and Kalderon, 2010*). Measurement of processing through Ci-155 levels and generation of repressor activity from *UAS-Ci* transgenes in wing discs previously showed that processing was absent only when the zinc finger and CORD domains were both removed (*Zhou and Kalderon, 2010*). To test whether Cos2-binding domains might be responsible for inhibiting Ci-155 activity we generated *ci* alleles lacking CDN, CORD or both regions. There are no known alterations to the zinc-finger region that affect Cos2 binding without compromising DNA binding and hence transcriptional activity of Ci-155.

We induced *pka* and *cos2* clones in wing discs expressing only CiΔCORD, CiΔCDN or CiΔCDNΔCORD. The level of *ptc-lacZ* in *pka* mutant clones was similar for wild-type Ci and CiΔCDN but it was significantly higher for CiΔCORD and CiΔCDNΔCORD; it was also higher for CiΔCORD than Ci-WT expressed from a *gCi* transgene (*Figure 6A–F,K*). These results indicate that the presence of the CORD domain reduces Ci-155 activity when Ci-155 is not processed in a *pka* mutant clone, while the CDN domain appears to have no impact on Ci-155 activity. By contrast, *ptc-lacZ* levels in *cos2* mutant clones were very similar for CiΔCORD, CiΔCDN, CiΔCORDΔCDN, and Ci-WT (*Figure 6G–K*). The simplest interpretation of these results is that Cos2 inhibits Ci-155 by binding to the CORD domain and that deletion of either the CDN or CORD domain does not affect any significant Ci-155 property other than binding to Cos2. Thus, in clones where Ci-155 is not processed the activity of Ci-155 is increased by loss of either Cos2 or the CORD domain but loss of the CORD domain cannot activate Ci-155 further in a *cos2* mutant clone. Moreover, the greater activity of Ci lacking the CORD domain in *pka* clones than in *cos2* clones shows that Ci-155 activation by loss of PKA activity and loss of Cos2-CORD binding can be additive.

In wing discs with no additional alterations, CiΔCORD, CiΔCDN, and CiΔCORDΔCDN all supported a near-normal Ci-155 profile, indicating substantially normal regulation of Ci-155 processing, and strong *ptc-lacZ* expression confined to the AP border (*Figure 7A–D*). There was a slight

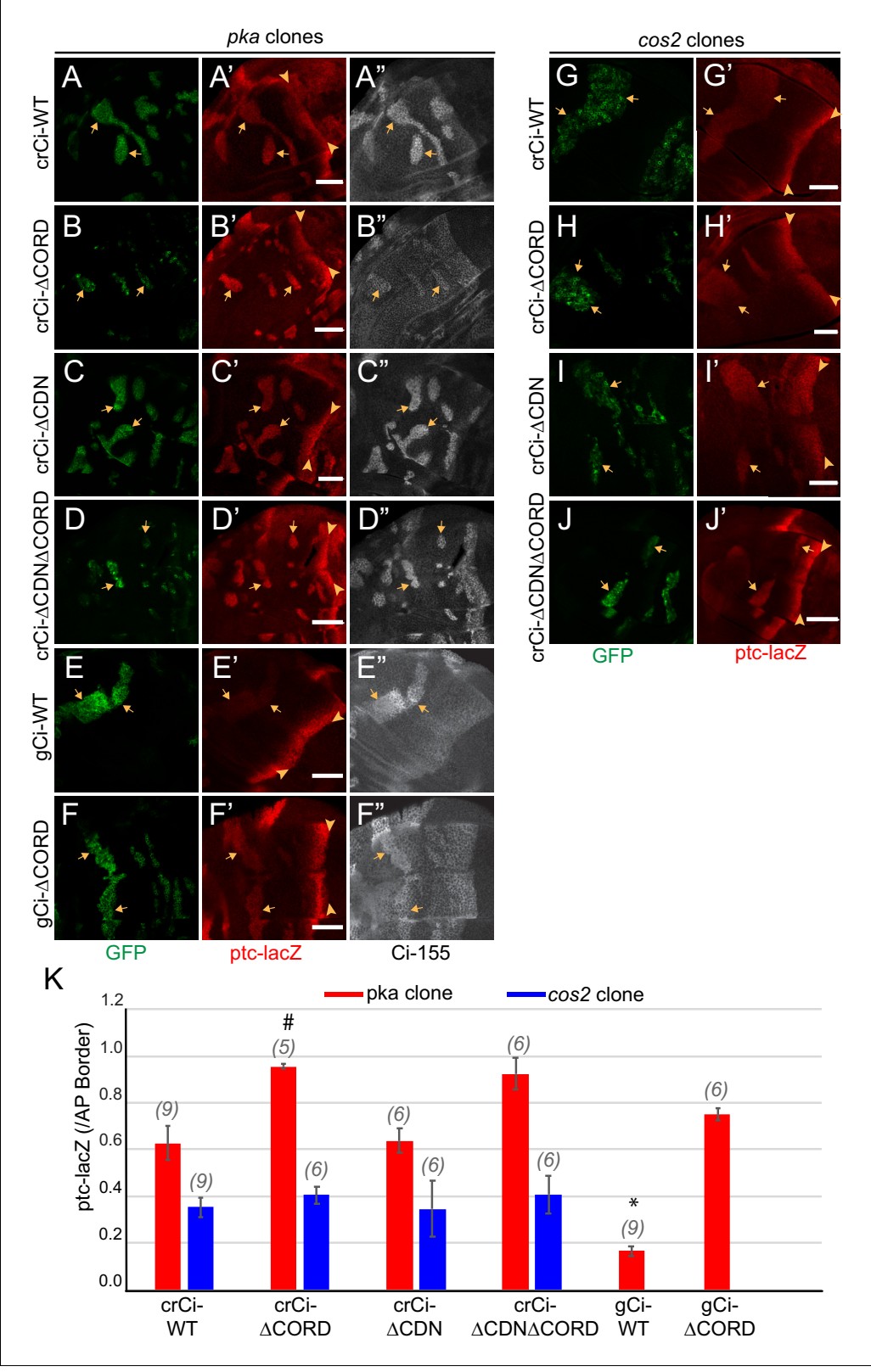

**Figure 6.** Cos2 reduces Ci-155 activity by binding to the CORD region. (A–J) Wing discs from animals with one copy of the designated *ci* transgenes and alleles (together with *ci^94*) with clones (GFP, green, arrows) that lack (A–F) *pka* activity or (G–J) *cos2* activity (arrowheads indicate AP border), showing (A'–J') *ptc-lacZ* (red) and (A"–F") Ci-155 (gray-scale). (A"–F") Ci-155 levels were increased relative to neighboring anterior territory for all Ci proteins, *Figure 6 continued on next page*

*Figure 6 continued*

but the increase was relatively small for (**B″**) Ci-ΔCORD, suggesting that processing outside the clones may be inefficient. By contrast, a large change was observed for Ci-ΔCDNΔCORD, suggesting very efficient processing. Scale bars are 40 μm. (**K**) Average intensity of *ptc-lacZ* in *pka* clones (red) and in *cos2* clones (blue), as a fraction of AP border levels. Mean and SEM shown. Significant differences between values for a given genotype compared to those for *crCi-WT*, calculated by paired t-tests, are indicated for p<0.001 (*) and p<0.05 (#). Additionally, *ptc-lacZ* was significantly increased for *gCi-ΔCORD* versus *gCi-WT* in *pka* mutant clones (p<0.0001).

The online version of this article includes the following source data for figure 6:

**Source data 1.** Numerical data for graphs in *Figure 6*.

enhancement of anterior Ci-155 levels for CiΔCORD, which was also evident in a *pka* heterozygous background (*Figure 6B*) and in a Su(fu) mutant background (*Figure 7—figure supplement 1A,C*). That may indicate a mild processing deficit. However, there was a very strong contrast between low anterior and high AP border Ci-155 levels of CiΔCORDΔCDN (*Figure 6D*; *Figure 7D*), supporting previous evidence that Ci-155 lacking both these Cos2-binding domains is processed very efficiently, perhaps even more efficiently than wild-type Ci, and that Hh blocks processing efficiently (*Zhou and Kalderon, 2010*). The experiments reported here, using Ci variants expressed at physiological levels, revealed a dependence on the Cos2-binding CORD domain for inhibiting Ci-155 activity that is not observed for Ci-155 processing or regulation of processing by Hh.

The absence of ectopic anterior *ptc-lacZ* in wing discs expressing CiΔCORD (*Figure 7B,D*) suggests that loss of Cos2-CORD association only leads to Ci-155 activity when Ci-155 processing is also inhibited. To test this hypothesis further, we created an allele expressing a processing-resistant Ci variant (S849A) that also lacked the CORD domain. We found that, unlike CiΔCORD, Ci-S849AΔCORD in combination with *ci*[94] resulted in wing discs with expanded anterior compartments and ectopic *ptc-lacZ* throughout the anterior (*Figure 7B,E*). Ectopic *ptc-lacZ* was much stronger than observed for Ci-S849A and was also evident cell autonomously in clones lacking a wild-type Ci transgene within wing discs expressing Ci-S849AΔCORD (*Figure 7K*). These results confirm that the CORD domain, which is only known to interact with Cos2, reduces Ci-155 activity when Ci-155 is not processed. Moreover, the observations that induction of *ptc-lacZ* in response to loss of PKA, Cos2 or the CORD domain depends on the dose of *ci* and protecting Ci-155 from processing are consistent with the idea that only Su(fu)-free Ci-155 is subject to inhibition by PKA and by Cos2 binding to the CORD domain.

## Additional CORD domain contributions

If the CORD domain serves only to permit Ci-155 inhibition by binding to Cos2, it might be expected that CiΔCORD either has the same activity at the AP border as wild-type Ci, or perhaps greater activity if Hh does not normally fully oppose Cos2-CORD interactions at the AP border. In fact, CiΔCORD (and CiΔCDNΔCORD) supported normal levels of *ptc-lacZ* but reduced En induction (*Figure 7A,B,D,F,G,I*). Loss of Su(fu) did not restore robust En expression (*Figure 7—figure supplement 1B,D*) and induction of *ptc-lacZ* was much reduced by loss of Fu kinase, just as for Ci-WT (*Figure 7—figure supplement 1E,F*). Wing discs lacking both Fu kinase and Su(fu) had strong *ptc-lacZ* but no En induction at the AP border for both CiΔCORD and Ci-WT (*Figure 7—figure supplement 1G,H*), consistent with an earlier report that Ci-155 activation by Fu operates substantially, but not entirely by antagonizing inhibition by Su(fu) (*Zhou and Kalderon, 2011*). We also found that CiΔCORD responded to activated GAP-Fu similarly to Ci-WT (*Figure 7—figure supplement 1I–L,N*). From these results, we speculate that the CORD domain may facilitate a facet of activation of Ci-155 by Fu that does not involve countering Su(fu) inhibition. For example, Fu activated by Hh at the AP border may not engage efficiently with Ci-155 complexes in the absence of the CORD domain, leading to a deficit in En induction, but excess GAP-Fu may largely compensate for that deficiency to produce similar activation of Ci-WT and CiΔCORD.

Surprisingly, despite its high constitutive activity, Ci-S849AΔCORD showed markedly lower induction of En at the AP border than CiΔCORD (*Figure 7J*), even though Ci-S849A induced En normally (*Figure 3B*). Since Ci-S849AΔCORD is neither processed nor inhibited by Cos2, it is presumably incompletely activated by Fu kinase activity, as hypothesized for CiΔCORD. The lesser induction of En when processing is fully inhibited suggests the possibility that those Ci-155 molecules spared

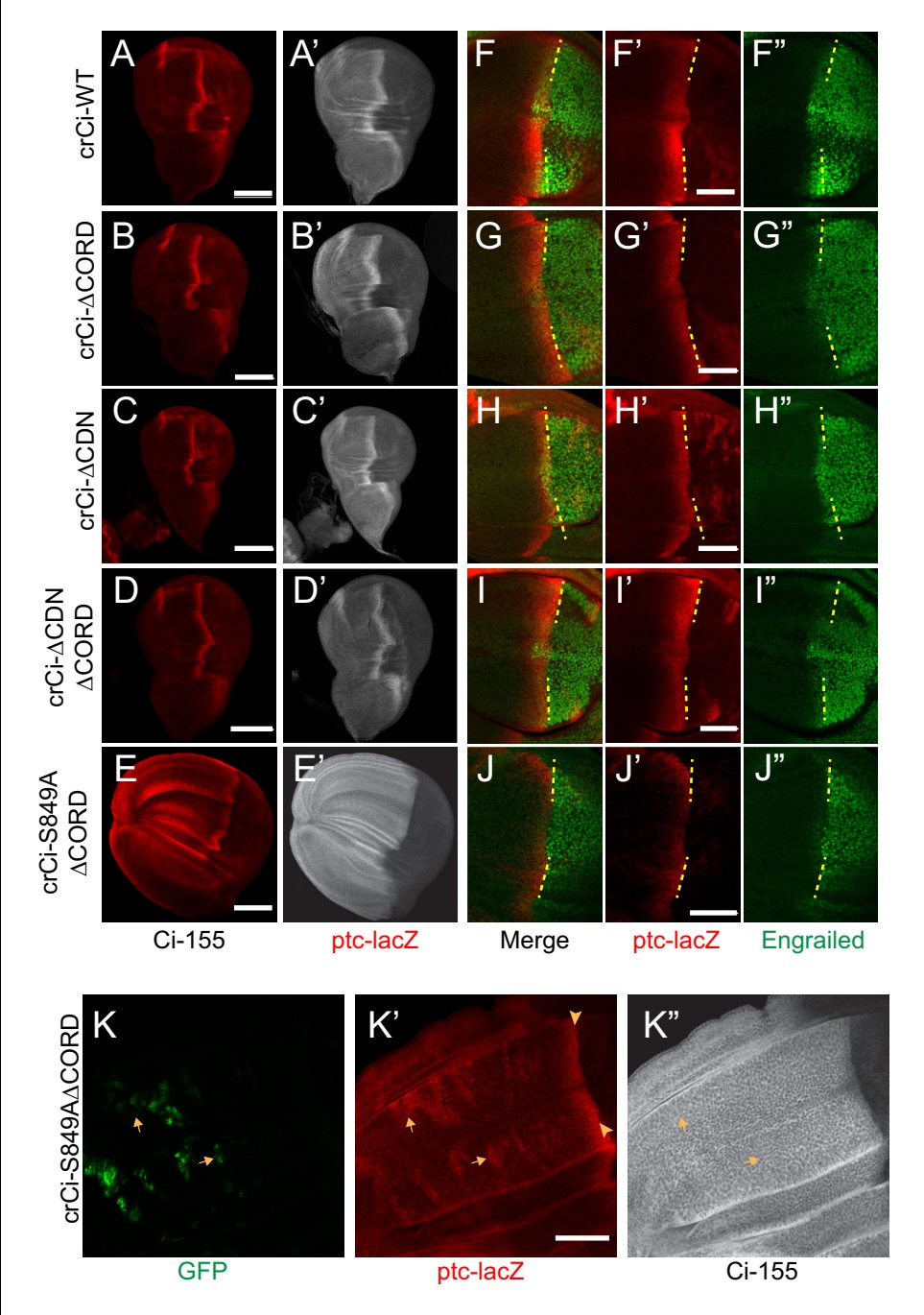

**Figure 7.** Loss of both Cos2 inhibition and processing combine to activate Ci-155. (**A–J**) Wing discs from animals with one copy of the designated *ci* alleles (together with *ci94*), have (**A–D**) no ectopic anterior *ptc-lacZ* (red) unless (**E**) both the CORD domain is removed and processing blocked (by the S849A alteration). (**A'–E'**) Ci-155 (gray-scale) in the same wing discs. (**F–J**) Anterior En (green) induction, revealed by marking the AP compartment boundary (yellow lines) with the posterior extent of *ptc-lacZ* (red), was reduced for Ci variants (**G, J**) lacking the CORD domain, (**J**) especially together with the S849A alteration. (**K**) Wing disc with anterior clones (GFP, green, yellow arrows) that have lost a second chromosome *gCi* transgene, leaving one copy of *crCi-S849AΔCORD* as the only source of Ci, showing (**K'**) *ptc-lacZ* induction in the clones (arrows) to levels similar to the AP border (arrowheads); (**K"**) Ci-155 (gray-scale) is uniformly high because of blocked processing. Scale bars are (**A–J**) 40 μm. See also *Figure 7—figure supplement 1*.

The online version of this article includes the following figure supplement(s) for figure 7:

*Figure 7 continued on next page*

**Figure supplement 1.** Reduced AP border activity of Ci lacking the CORD domain.

from processing but failing to engage with activated Fu might compete with activated Ci-155 and thereby limit Hh target gene induction.

## Discussion

Hh signaling in *Drosophila* and in mammals involves two key changes: inhibition of the proteolytic processing of Ci/Gli proteins to repressor forms, thereby also increasing full-length protein levels, and activation of full-length Ci/Gli proteins (*Figure 8*). The relative importance of repressor and activator, both of which can potentially regulate the same set of genes, varies in different mammalian tissues in part because of the Gli protein expressed (Gli3 is more efficiently converted to repressor than Gli2) and because Gli1 is itself a Hh target gene, acts only as an activator and therefore has a specialized amplification role (*Briscoe and Thérond, 2013*; *Kong et al., 2019*; *Liu, 2019*). In *Drosophila*, Ci is the only transcriptional effector of Hh signaling, allowing straightforward interrogation of the relative importance of regulation through altering the levels of Ci-75 repressor, latent Ci-155 activator and conversion of Ci-155 to a potent transcriptional activator. Moreover, wing disc development is perhaps the most demanding and easily perturbed patterning challenge for Hh signaling

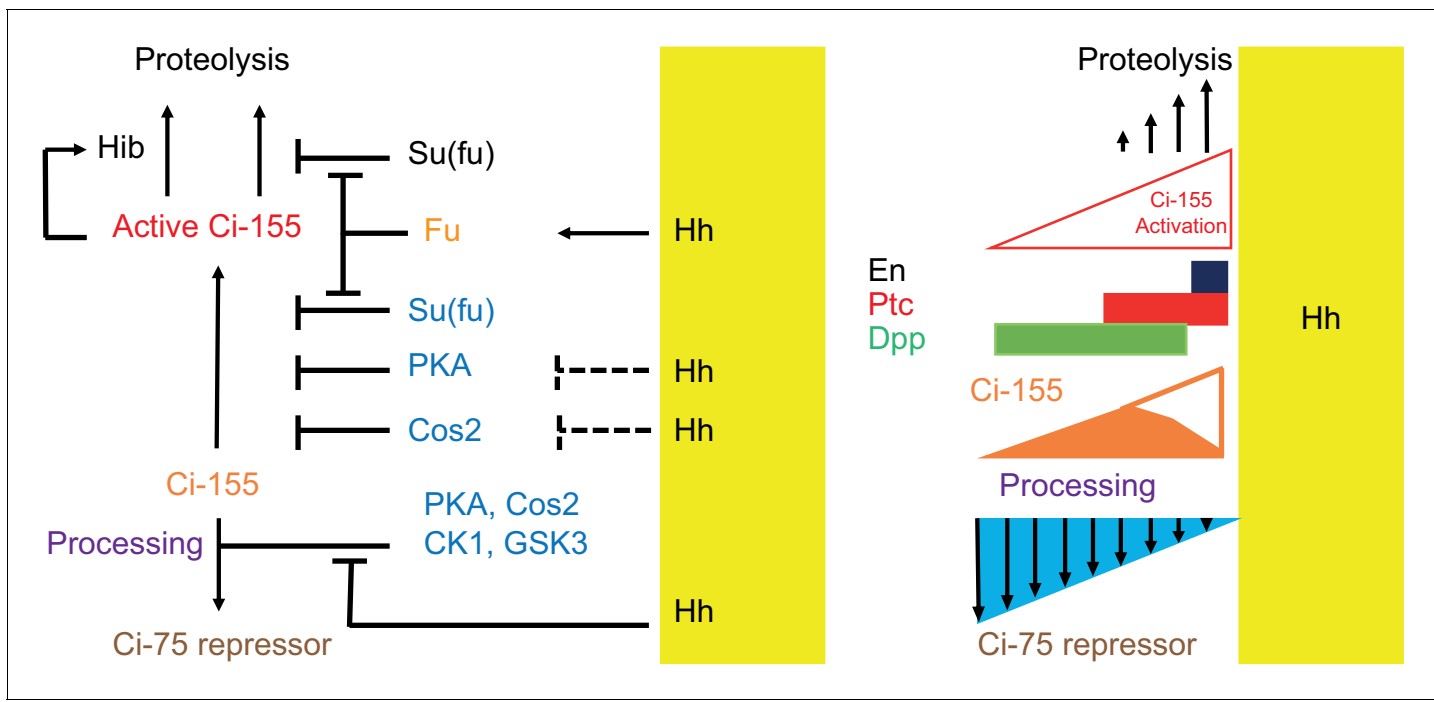

**Figure 8.** Summary of graded processing, activation, and proteolysis of Ci-155 and underlying mechanisms at the AP border. At the AP border, Hh (emanating from posterior, mustard yellow, territory) inhibits Ci-155 processing stimulated by Cos2, PKA, CK1, and GSK3, and activates Fu protein kinase activity to activate Ci-155, overcoming inhibition by Su(fu) and other factors. Hh also promotes a reduction of Ci-155 levels, most likely by promoting full Ci-155 proteolysis through induction of Rdx/Hib or reducing Su(fu) association. Here we used processing-resistant Ci variants to show that PKA and Cos2 (through binding the CORD domain on Ci-155) limit Ci-155 activity in anterior cells (left) and (right) to deduce the spatial profiles of Hh-stimulated Ci-155 reduction ('proteolysis', upward arrows) and inhibition of Ci-155 processing (downward arrows, blue triangle) at the AP border that underlie steady-state Ci-155 levels (brown). Graded target gene (En, Ptc, Dpp) activation is normally elicited by a combination of activated Ci-155 and Ci-75 repressor but was still observed when there was no regulation of Ci-155 processing, indicating that Ci-155 activation must be graded. Although graded Hh signaling was observed when Ci-155 processing is not regulated, Ci-75 repressor must be present in anterior cells to prevent ectopic *dpp* expression and inhibition of processing was shown to be important for activated Fu to induce high levels of *ptc* expression. Thus, Hh normally elicits graded inhibition of Ci-155 processing and graded activation of full-length Ci-155 but the activation gradient can suffice provided there is some repressor in anterior cells and Ci-155 is substantially spared from processing at the AP border.

in *Drosophila* and therefore suitable for dissecting essential regulatory influences that support dose-dependent responses. It is therefore remarkable that we found that regulation of Ci-155 processing is not essential for major manifestations of Hh morphogen action in wing discs.

## Evidence that regulation of Ci-155 processing is not essential for Hh morphogen action

Processing of Ci-155 is initiated by phosphorylation of three PKA sites ('P1-3') and involves the creation of a Slimb-SCF complex binding site that includes phosphorylated S849. It has previously been shown that alteration of the PKA sites (P1-3A) or S849 (S849A) abrogates Slimb binding in vitro, Ci-75 production detected by Western blot of embryo extracts expressing HA-tagged transgenes and all Ci-75 repressor activity, assayed by *hh-lacZ* repression in *smo* mutant clones expressing *ci* transgenes in the posterior compartment of wing discs (*Méthot and Basler, 2000*; *Price and Kalderon, 1999*; *Price and Kalderon, 2002*; *Smelkinson and Kalderon, 2006*; *Smelkinson et al., 2007*). Moreover, we found here that Ci-P(1-3)A and Ci-S849A expressed at physiological levels produced high levels of Ci-155 throughout the anterior with no elevation at the AP border, and that *dpp-lacZ* was ectopically expressed in anterior cells, as expected if no Ci repressor is present (*Méthot and Basler, 1999*). Thus, the absence of processing for Ci-P(1-3)A and Ci-S849A has been firmly established.

We found that normal patterns of induction of the Hh target genes *ptc-lacZ* and En at the AP border were supported by a Ci variant that cannot be processed. We tested only one copy of the *ci-S849A* and *ci-P(1-3)A* alleles, so it remains possible that two copies might impair patterning. We also found that one *crCi-S849A* allele together with a genomic *gCi-S849A* transgene and the constitutive repressor allele *ci^Ce* produced adults with normally patterned wings. The result shows that wing patterning by Hh does not require regulation of the level of either repressor or full-length Ci protein by processing.

Animals lacking both Fu kinase and Su(fu) also develop normal wings, although late third instar wing discs do lack anterior En induction (*Ohlmeyer and Kalderon, 1998*; *Préat, 1992*; *Zhou and Kalderon, 2011*). Thus, regulation of Ci-155 processing and the most prominent regulator of Ci-155 activation, Fu kinase activity, are each largely dispensable for Hh morphogen action, suggesting that each graded patterning mechanism can suffice in the absence of the other. The spatial morphogen action of Hh is aided by the negative feedback loop of *ptc* transcriptional induction leading to increased Hh sequestration by Ptc protein (*Chen and Struhl, 1996*). That feature diminishes Hh spread through cells with hyper-sensitive signal transduction and increases spread through cells of reduced sensitivity, potentially accommodating limited deficiencies in signal transduction due to the loss of one major mode of Ci activity regulation.

The profile of Hh inhibition of Ci-155 processing had not previously been observed or deduced because wild-type Ci-155 is also subject to Hh-stimulated degradation and potentially other changes that also affect Ci-155 levels. Here, we have derived a clear profile of Hh-stimulated processes that lead to reduced Ci-155 by examining Ci variants that are not subject to processing. The Hh-stimulated decline in Ci-155 levels begins at a location where *ptc-lacZ* induction is roughly half-maximal and is roughly linear, resulting in a reduction of over twofold by the compartment boundary (*Figure 2H", I", K*; *Figure 8*). Although the mechanisms contributing to Hh-stimulated reduction in Ci-155 are not fully understood (see below), they appear always to be in proportion to pathway activity. Since both Ci-P(1-3)A and Ci-S849A have the same pathway activity profiles as Ci-WT, measured by *ptc-lacZ* and En, we assume that the Ci-155 reduction profile observed directly for the processing-resistant variants is very similar for wild-type Ci. We therefore added the observed value of Ci-155 loss at each AP location to the observed Ci-155 profile of wild-type Ci to deduce the normal Ci-155 profile due to processing alone (*Figure 2L*; *Figure 8*). The inhibition of Ci-155 processing extended from a location slightly anterior to the edge of *ptc-lacZ* induction to the AP border in a clearly graded manner that, in isolation, would alter Ci-155 levels more than two-fold. Thus, we have derived the first clear visualization of graded inhibition of Ci-155 processing and of graded, pathway-stimulated, Ci-155 loss.

## Hh-stimulated Ci-155 reduction

Both the mechanism and the purpose of Hh-stimulated Ci-155 reduction at the AP border remain uncertain. It was initially suggested that Hh-stimulated Ci-155 reduction was due to the transcriptional induction of Rdx/Hib, which bound activated Ci-155 directly to promote its degradation and limit the magnitude of Hh target gene induction by the highest levels of Hh (*Kent et al., 2006*; *Zhang et al., 2009*; *Zhang et al., 2006*). However, other studies found that Ci-155 levels remained low in high Hh signaling territory even when Rdx/Hib activity was eliminated and that Rdx/Hib might influence Ci-155 indirectly via modulation of Su(fu) protein levels (*Liu et al., 2014*; *Seong et al., 2010*; *Seong and Ishii, 2013*).

Here, we specifically tested the contribution of direct targeting of Ci by Rdx/Hib for the first time in a physiological setting by using a Ci variant with multiple alterations to sites of Rdx/Hib association; those alterations had been shown to nearly eradicate direct down-regulation of Ci-155 by Hib E3 ligase complexes under synthetic conditions (*Zhang et al., 2009*). This Ci variant (Ci-S3-5) supported a normal pattern of Hh target gene induction in wing discs and the development of adults with normal wings. The Ci-155 profile was also very similar to wild-type Ci with robust Hh-stimulated Ci-155 reduction in the posterior part of the AP border. While the Ci variant may retain some residual Hib binding, our results suggest that the majority of Hh-promoted Ci-155 reduction is through mechanisms other than degradation due to direct binding of Rdx/Hib.

The complete absence of Su(fu) greatly reduces Ci-155 levels but not *ci* RNA levels throughout wing discs (*Ohlmeyer and Kalderon, 1998*), leading to the hypothesis that direct binding of Su(fu) to Ci-155 protects Ci-155 from degradation. It is also commonly speculated that pathway activation elicits Ci-Su(fu) dissociation, as suggested by studies of mammalian Hh signaling (*Humke et al., 2010*; *Tukachinsky et al., 2010*). Such dissociation, if stimulated in proportion to Fu activation, would promote Ci-155 degradation in proportion to Ci-155 activation without the necessary participation of a transcriptionally induced intermediate, such as Rdx/Hib (*Figure 8*). Consistent with this hypothesis, no reduction of Ci-155 levels close to the source of Hh was apparent in wing discs lacking Su(fu). The sensitivity of those measurements was, however, limited by the low Ci-155 levels throughout such wing discs. Whether Ci-155 activation by Fu does involve dissociation of Su(fu) and whether that contributes significantly to pathway-stimulated Ci-155 degradation remain to be thoroughly investigated.

Although the Rdx/Hib and Su(fu)-dependent proteolytic mechanisms outlined above are prominent candidates for mediating Hh-stimulated reduction of Ci-155 at the AP border, it is possible that transcriptional, RNA processing, or translational mechanisms are also involved. Initial studies of *ci* RNA and a *lacZ* enhancer trap of the *ci* locus (*ci-lacZ*) suggested that third instar larvae have spatially uniform anterior *ci* transcription and RNA (*Eaton and Kornberg, 1990*; *Ohlmeyer and Kalderon, 1998*). *ci-lacZ* was, however, seen to be markedly lower in AP border regions 30 hr after pupariation, possibly resulting from transcriptional repression of *ci* by En, which is itself induced in anterior cells only in late third instar larvae (*Blair, 1992*). Other studies have shown that the pattern of *ci* RNA splicing and overall RNA levels can be selectively altered by reduced activity of the exon-junction complex or the splicing factor, Srp54, suggesting the potential to regulate *ci* RNA processing (*Garcia-Garcia et al., 2017*).

## Ci-155 activation by Fu

We found that artificially activated Fu (GAP-Fu) can activate processing-resistant Ci in anterior, Hh-free territory as effectively as normal Fu activity at the AP border. Wild-type Ci activated by GAP-Fu induced roughly two-fold lower levels of *ptc-lacZ*, and barely induced *ptc-lacZ* at all if it was produced at slightly lower levels by a *gCi* transgene rather than a *ci* allele. These results are consistent with the simple idea that more activated Ci-155 molecules collectively induce transcription more strongly. However, the steady-state level of Ci-155 in cells with synthetically activated Fu was not significantly higher for processing-resistant variants than for wild-type Ci, presumably because of robust Ci-155 degradation in response to high pathway activity. An analogous circumstance is apparent in posterior regions of the AP border: Ci-155 processing is largely inhibited, Fu kinase and Ci-155 are strongly activated but Ci-155 levels are similar to those in anterior cells because of robust Hh-stimulated Ci-155 reduction. The GAP-Fu experiment reports that high pathway activity causes a high rate of Ci-155 loss and that high pathway activity can only be maintained if there is an adequate, constant

supply of fresh Ci-155 protected from processing. This imposed requirement might be the major purpose of Hh-promoted Ci-155 reduction at the AP border, rather than modulating the profile of the Hh signaling gradient. Under this arrangement, cells will continue to express high-level Hh target genes only when constantly stimulated. The arrangement also allows for the possibility of modulating pathway activity through both the degree of Fu activation and the rate of supply of Ci-155 that is protected from processing.

## Ci-155 activity regulation by PKA and Cos2

We used the processing-deficient variant Ci-P(1-3A) to show that genetic removal of PKA or Cos2 substantially increased Ci-155 activity in the absence of Hh, providing evidence that both PKA and Cos2 inhibit Ci-155 activation in addition to their well-established roles of promoting Ci-155 processing (*Figure 8*). Earlier tests concerning the role of PKA generally reached the same conclusion but were subject to a number of caveats (*Smelkinson et al., 2007*; *Wang et al., 1999*). The findings reported here supersede those conclusions because physiological expression of Ci-155 variants was assayed in normal locations. They also showed that the magnitude of inhibition by Cos2 and PKA was substantial and allowed some exploration of the mechanisms involved.

We found that removal of the CORD domain of Ci conferred significantly higher activity on a processing-resistant Ci variant and increased the response of otherwise normal Ci to loss of PKA, but not to loss of Cos2. These observations are consistent with the hypothesis that the CORD domain is the major mediator of the inhibitory action of Cos2. By contrast, deletion of the CDN Cos2-binding domain did not alter the activity of processing-resistant Ci. Ci-155 processing remained efficient in the absence of both CDN and CORD domains, confirming a previous deduction from *UAS-Ci* transgenes that Cos2 binding to the zinc finger domain of Ci can suffice to promote processing (*Zhou and Kalderon, 2011*). Removal of the CORD domain reduced En induction at the AP border and we hypothesize that this might result from a deficiency in targeting activated Fu to Ci. Thus, although Ci-155 has three domains that can bind to Cos2, it appears that they do not contribute equally to regulate Ci-155 processing, inhibition and activation.

We did not resolve how PKA inhibits Ci-155. The finding that loss of PKA increased the activity of Ci-P(1-3) shows that the PKA sites used to direct processing (P1-3) are not essential targets for PKA to inhibit Ci-155. Ci-155 includes two additional consensus sites at residues 962 and 1006. In earlier studies using multiple *UAS-Ci* transgenes at a variety of genomic locations, Ci variants lacking all five PKA sites (P1-5A) were found to be more active than those lacking just sites P1-3 (*Price and Kalderon, 1999*). However, the relative levels of *ci* transgene expression were not measured in that study and all were likely higher than physiological levels. In mouse studies, evidence was provided, albeit with non-physiological expression levels, that alteration of PKA sites in Gli2 analogous to residues 962 and 1006 in Ci-155 increased Gli2 activity (*Niewiadomski et al., 2014*). We were unable to recover a *crCi* allele encoding a variant with all five PKA sites altered. We were similarly unable to recover variants with processing-resistant alterations together with Su(fu)-binding site alterations, and the processing- resistant variant with a CORD domain deletion was also difficult to recover and propagate. We speculate that these difficulties may all derive from a shared characteristic of constitutively high activity, providing a hint that PKA sites 962 and 1006 might be important to restrain Ci-155 activity. However, both these sites are within the CORD domain and Ci lacking the CORD domain is more strongly activated by loss of PKA than by loss of Cos2, indicating that PKA inhibition does not require PKA sites 4 and 5. There may, of course, be more than one target through which PKA inhibits Ci-155 activation, including the possibility that PKA acts separately through sites 1–3 and 4–5.

Another unresolved issue is to what extent Hh signaling at the AP border antagonizes the inhibitory influences of Cos2 and PKA. When Hh signals, Ci-155 processing is reduced primarily through partial dissociation of Cos2-Ci complexes (*Li et al., 2014*; *Ranieri et al., 2014*) and this processing inhibition occurs even in the absence of Fu kinase activity (*Ohlmeyer and Kalderon, 1998*; *Smelkinson and Kalderon, 2006*). It is not clear what degree of dissociation is elicited at the AP border or whether Cos2-CORD interactions might be altered within intact Cos2-Ci complexes to relieve Cos2 inhibition. Ci-P(1-3)A induced no *ptc-lacZ* in anterior cells, low *ptc-lacZ* levels at the AP border of Fu-kinase deficient discs and significantly higher *ptc-lacZ* levels in *cos2* and *pka* mutant clones. We can therefore deduce that in the absence of Fu kinase there may be some reduction of inhibition by Cos2 and PKA at the AP border, leading to low *ptc-lacZ* induction, but the reduction is

much less than from complete elimination of Cos2 or PKA activities. It is possible that Hh additionally counters inhibition by Cos2 or PKA through Fu activation. Indeed, anterior En induction at the AP border requires Fu activity even in the complete absence of Su(fu), showing that Fu opposes Ci-155 inhibition by factors other than Su(fu) (*Zhou and Kalderon, 2011*). Cos2 and PKA are the only other known inhibitory factors.

In summary, at the AP border of wing discs, Hh inhibits Ci-155 processing, activates full-length Ci-155 and promotes reduction of Ci-155, most likely substantially through proteolytic degradation (*Figure 8*). Processing-resistant Ci variants revealed the profiles of Hh-promoted Ci-155 reduction and Ci-155 processing inhibition (*Figure 2K,L* and *Figure 8*), and showed that Hh can pattern wing discs and wings normally in the absence of regulated processing. Ci variants lacking Rdx/Hib-binding sites showed that Ci-155 reduction likely depends on Hh-stimulated processes other than direct binding to the transcriptionally induced component of an E3 ubiquitin ligase, plausibly involving protection from degradation by Su(fu) association (*Figure 8*). We also found that Ci-155 that is not subject to processing is substantially inhibited by PKA and by association with Cos2 through the CORD domain in addition to Su(fu), and that activation by Fu only elicits strong induction of Hh target genes if there is a continued ample supply of Ci-155 protected from processing.

# Materials and methods

## Key resources table

| Reagent type (species) or resource | Designation | Source or reference | Identifiers | Additional information |
|---|---|---|---|---|
| Gene (*Drosophila melanogaster*) | Ci | Flybase ID: FBgn0004859 | CG2125 | |
| Gene (*Drosophila melanogaster*) | Cos2 | Flybase ID: FBgn0000352 | CG1708 | |
| Gene (*Drosophila melanogaster*) | PKA | Flybase ID: FBgn0000273 | CG4379 | |
| Gene (*Drosophila melanogaster*) | Fused | Flybase ID: FBgn0001079 | CG6551 | |
| Gene (*Drosophila melanogaster*) | Suppressor of Fused | Flybase ID: FBgn0005355 | CG6054 | |
| Genetic reagent (*Drosophila melanogaster*) | hs-flp | PMID:7867064 | FBti0002738 | hsp70-driven Flp recombinase on X |
| Genetic reagent (*Drosophila melanogaster*) | ci[94] | PMID:7705626 PMID:10102270 | FBal0045443 | 5 kb deletion removing promoter and first exon |
| Genetic reagent (*Drosophila melanogaster*) | ci[Ce] | PMID:10102270 | ci[Ce2] FBal0001657 | 8 bp deletion that is expected to result in a truncation of the protein at amino acid residue 975 |
| Genetic reagent (*Drosophila melanogaster*) | Dp[y⁺] | PMID:10102270 | *Dp(1;4)1021[y⁺] sv[spa-pol]* FBab0003151 | |
| Genetic reagent (*Drosophila melanogaster*) | Su(fu)[LP] | PMID:1468628 | FBal0016296 | Amorphic 1.5 kb deletion extending into neighboring *kar* gene |
| Genetic reagent (*Drosophila melanogaster*) | pka-C1[H2] | PMID:8391504 | FBal0033960 | G203D alteration to key kinase domain residue |
| Genetic reagent (*Drosophila melanogaster*) | smo[2] | PMID:15592457 | FBal0015765 | Behaves as a null |
| Genetic reagent (*Drosophila melanogaster*) | FRT 42D P[Smo⁺] | PMID:10102270 | P[Smo⁺, hsp70-GFP] FBtp0012072 | Fully rescues loss of *smo* function |
| Genetic reagent (*Drosophila melanogaster*) | FRT 42D cos2[2] | PMID:11090136 | FBal0001772 | To generate loss-of-function cos2 clones |
| Genetic reagent (*Drosophila melanogaster*) | fu[mH63] | PMID:8846897 | FBal0120493 | G203D loss of kinase activity |

*Continued on next page*

*Continued*

| Reagent type (species) or resource | Designation | Source or reference | Identifiers | Additional information |
|---|---|---|---|---|
| Genetic reagent (*Drosophila melanogaster*) | tub-GAL80 FRT 40A | BDSC BL-5192 | | For MARCM clones on 2L |
| Genetic reagent (*Drosophila melanogaster*) | FRT 42D P[Ci$^+$] tub-GAL80 | PMID:10102270 | | P[Ci+] 16 kb segment rescues *ci* null in stock for 2R MARCM clones |
| Genetic reagent (*Drosophila melanogaster*) | C765 > Gal4 | Flybase ID: FBti0002765 | | Spatially uniform wing disc GAL4 driver |
| Genetic reagent (*Drosophila melanogaster*) | UAS-GAP-Fu | PMID:17658259 | FBal0284373 | Fu coding sequence with Myristoylation sequence from hGAP43 at N-terminus and CFP at C-terminus |
| Genetic reagent (*Drosophila melanogaster*) | ptc-lacZ | PMID:8898207 | P[ptcA-lacZ] FBal0047864 | 10.8 kb ptc promoter driving lacZ |
| Antibody | Anti-Ci-155 (rat monoclonal) | DSHB | AB_2109711 | (1:3) |
| Antibody | Anti-beta-galactosidase (rabbit polyclonal) | MP Biomedicals | AB_2334934 | (1:10,000) |
| Antibody | Anti-Engrailed (mouse monoclonal) | DSHB | AB_528224 | (1:5) |
| Antibody | AlexaFluor 488, 546, 594, 647 | Thermofisher Scientific | Anti-rabbit, Anti-mouse Anti-Rat | (1:1000) |
| Recombinant DNA reagent | pCFD4 | Mann Lab | Addgene: 83954 | Gibson cloning of gRNA |
| Recombinant DNA reagent | Bluescript genomic Cubitus interruptus | Basler Lab | | |
| Recombinant DNA reagent | att-Pacman Expression Vector | DGRC | | |
| Chemical compound, drug | Normal Goat Serum | Jackson Immunoresearch laboratories | RRID:AB_2336990 | |
| Chemical compound, drug | Aqua Polymount | PolySciences | CN: 18606–20 | |
| Commercial assay or kit | Gibson Assembly | New England Biolabs | CN: E5510S | |
| Commercial assay or kit | PfuUltraII Fusion HS DNA polymerase | Agilent Technologies | CN: 600670 | |
| Commercial assay or kit | Zero Blunt Topo cloning vector | Invitrogen | CN: K270020 | |
| Strain, strain background (*Escherichia coli*) | Transformax EPI 300 Electrocompetent *E. coli* | Epicentre Now lucigen | CN: EC300110 | Electro-competent cells |
| Strain, strain background (*Escherichia coli*) | One Shot TOP10 Chemically Competent *E. coli* | Thermofisher Scientific | CN: C4040-10 | Chemically Competent cells |
| Software, algorithm | Image J | NIH Bethesda Maryland | | |
| Software, algorithm | A Plasmid Editor (APE) | | | |

### Genomic *ci* cloning

Genomic transgenes were created by cloning the entire 16 kb genomic *ci* region from a Bluescript-SK (BSK) vector (provided by Dr. K. Basler; *Méthot and Basler, 1999*) into an *att-Pacman* Expression vector (DGRC). To facilitate mutagenesis, the 16 kb fragment was first separated into two parts. The region including the promoter, first exon and part of the first intron ('Ci fragment 2') was cloned as a

BamHI-NheI fragment into BSK cut with BamHI and XbaI to create BSK-CiF2. The complementary NheI-KpnI fragment containing all other exons and the 3' UTR ('Ci Fragment 1') was cloned into BSK cut with SpeI and KpnI to create BSK-CiF1. BSK-CiF2 was cut with NotI and Bsp1201 to clone the whole CiF2 fragment into the P[acman]-CmR vector cut with NotI, so that RsrII and PmeI vector sites were downstream of *ci* first intron sequences in RP-CiF2. CiF1 was amplified from BSK-CiF1 by long-range PCR using PfuUltraII Fusion HS DNA polymerase (Agilent Technologies), adding RsrII and PmeI at either end and cloning the product into a Zero Blunt Topo cloning vector (Invitrogen). The RsrII-PmeI fragment was then cloned into RP-CiF2 cut with the same enzyme to create the final Pacman vector containing the entire 16 kb genomic ci DNA. The 28 kb *gCi attPacman* transgene was then inserted at the *att ZH-86Fb* landing site at cytological location 86F8 (Rainbow Transgenic Services).

## Cloning for generating CRISPR alleles
### First round of CRISPR
A 5kb *mini-white* gene from the *attPacman* construct was cloned into the first intron of 'Ci Fragment 1' with the enzyme AalI. The PAM sites associated with guide RNA 1 (TGG->TGA) and guide RNA 2 (TGG->TTG) were mutated on 'Ci-Fragment 1' in the Ci first intron. guide RNA 1 TCACCCAAAAA TCTCGTATT and guide RNA 2 ATATATATACAAGAGTTCCT were cloned in pU6 chiRNA vectors separately. The donor template, guide RNA 1, and guide RNA two were then co-injected into fly embryos (*wlig4; attp40 [nos-Cas9]/Cyo*). Flies and guide RNA vectors were obtained from the Mann Lab and injections were carried out using Rainbow Transgenic Services. The injected flies were crossed to *yw hs-flp; Sp/Cyo; TM2/TM6B; Dp[y+]/Dp[y+]* flies (*Dp[y$^+$]* is used throughout as an abbreviation for *Dp(1;4)1021[y$^+$]sv$^{spa-pol}$*) and progeny screened for male flies that were white$^+$. The transformants were balanced and further genotyped to confirm correct placement of the *mini-white* gene (reverse coding orientation compared to *ci*) in the intron. The *ci-[w$^+$]* flies (4$^{th}$ chromosome) were used to create a stock, *wlig4; attp40 [nos-Cas9]/Cyo; ci-[w$^+$]/ci-[w$^+$]*.

### Second round of CRISPR
'Ci Fragment 1' was repurposed as donor construct by adding 500 bp extra on the 3'UTR region to create a 1.1 Kb homology region outside of guide RNA 3 and 2 kb homology region outside of guide RNA 4. PAM sites were altered on the donor construct for guide RNA 3 (GGG->CCG) and guide RNA 4 (CGG->CAG). guide RNA 3 (GGGCTTACGCCGGTATTAG) and guide RNA 4 (GC TTTGGGTGTAGGAGCGTC) were cloned into a dual U6 (1+three promoter) expression construct pCFD4 provided by the Mann lab using Gibson assembly (New England Biolabs). The donor construct and the guide RNA construct were injected into *wlig4; attp40 [nos-Cas9]/Cyo; ci-[w$^+$]/ci-[w$^+$]* embryos. Surviving adults were crossed to *yw hs-flp; Sp/Cyo; TM2/TM6B; Dp[y+]/Dp[y+]* flies. Male *crCi/Dp[y+]* 'transformants' were identified by white eyes, amplified into suitable stocks and genotyped for sequences encoding Flag and HA tags upstream and downstream of *ci* coding sequence, respectively. Balanced *ci* alleles were further genotyped to confirm the mutation of interest.

## Donor template cloning
For crCi-WT, ΔCORD, P(1-3)A, S849A, S849AΔCORD, Δ1270–1370, ΔCDN, ΔCDNΔCORD plasmid design was developed using APE software. Overlapping primer PCR reactions were used to add, mutate, and delete regions on Ci with PfuUltraII Fusion HS DNA polymerase (Agilent Technologies). PCR products were introduced into the Zero Blunt Topo cloning vector (Invitrogen). The alterations in Ci were then re-introduced from the Zero Blunt Topo cloning Vector into the BSK-F1 Donor construct using compatible enzymes or Gibson Assembly (New England Biolabs). The final constructs were fully sequenced (Genewiz).

## *Drosophila* stocks
*Drosophila* stocks were maintained on standard cornmeal/molasses/agar medium at room temperature.

Females of the genotype *yw hs-flp; ptc-lacZ/TM6B, Tb; ci$^{94}$/Dp[y$^+$]* were crossed to *yw hs-flp; Sp/Cyo; gCi-WT/ΔCORD/S849A; ci$^{94}$/Dp[y$^+$]* males, selecting third instar larval progeny lacking *y$^+$* and *Tb* to obtain wing discs with third chromosome transgenes as the only source of Ci.

Females of the genotype *yw hs-flp; ptc-lacZ/TM6B, Tb; ci^94^/Dp[y^+^]* were crossed to *yw hs-flp; Sp/Cyo; crCi-X/Dp[y^+^]* males, selecting third instar larval progeny lacking *y^+^* and *Tb* to obtain wing discs with a single constructed *crCi* allele as the only source of Ci.

Females of the genotype *yw hs-flp; Su(fu)^LP^ ptc-lacZ/TM6B, Tb; ci^94^/Dp[y^+^]* were crossed to *yw hs-flp; Sp/Cyo; Su(fu)^LP^/TM6B, Tb; crCi-X/Dp[y^+^]* males, selecting third instar larval progeny lacking *y^+^* and *Tb* to obtain wing discs with a single constructed *crCi* allele as the only source of Ci in a *Su(fu)* null background.

Females of the genotype ('2L') *yw hs-flp UAS-GFP; tub-Gal80 FRT40A/Cyo; C765-GAL4 ptc-lacZ/TM6B, Tb; ci^94^/Dp[y^+^]* were crossed to males of the genotype *yw hs-flp; pka-C1^H2^ FRT40A/Cyo; gCi-WT/ΔCORD/S849A/TM6B, Tb; ci^94^/Dp[y^+^]* or *yw hs-flp; pka-C1^H2^ FRT40A/Cyo; crCi-X/Dp[y^+^]*, selecting third instar larval progeny lacking *y^+^* and *Tb* to obtain wing discs with a single constructed *crCi* allele as the only source of Ci and GFP-marked *pka* mutant clones.

Females of the genotype ('2b') *yw hs-flp UAS-GFP; FRT42D P[Ci^+^] tub-Gal80/Cyo; C765-GAL4 ptc-lacZ/TM6B, Tb; ci^94^/Dp[y^+^]* were crossed to males of the genotype *yw hs-flp; FRT42D/Cyo; crCi-X/Dp[y^+^]*, selecting third instar larval progeny lacking *y^+^* and *Tb* to obtain wing discs with a single constructed *crCi* allele as the only source of Ci in GFP-marked clones lacking *P[Ci^+^]* with neighboring cells including *P[Ci^+^]*.

Females of the genotype ('2b') *yw hs-flp UAS-GFP; FRT42D P[Ci^+^] tub-Gal80/Cyo; C765-GAL4 ptc-lacZ/TM6B, Tb; ci^94^/Dp[y^+^]* were crossed to males of the genotype *yw hs-flp; FRT42D cos2^2^/Cyo; gCi-WT/ΔCORD/S849A/TM6B, Tb; ci^94^/Dp[y^+^]* or *yw hs-flp; FRT42D cos2^2^/Cyo; crCi-X/Dp[y^+^]*, selecting third instar larval progeny lacking *y^+^* and *Tb* to obtain wing discs with a single constructed *crCi* allele as the only source of Ci in GFP-marked clones lacking *cos2* activity and *P[Ci^+^]* with neighboring cells expressing *P[Ci^+^]*.

Females of the genotype ('2b') *yw hs-flp UAS-GFP; FRT42D P[Ci^+^] tub-Gal80/Cyo; C765-GAL4 ptc-lacZ/TM6B, Tb; ci^94^/Dp[y^+^]* were crossed to males of the genotype *yw hs-flp; smo^2^ FRT42D UAS-GAP-Fu/Cyo; crCi-X/Dp[y^+^]*, selecting third instar larval progeny lacking *y^+^* and *Tb* to obtain wing discs with a single constructed *crCi* allele as the only source of Ci in GFP-marked clones expressing GAP-Fu and lacking *P[Ci^+^]* with neighboring cells expressing *P[Ci^+^]*.

Females of the genotype ('2R') *yw hs-flp UAS-GFP; smo^2^ FRT42D P[Smo^+^] tub-Gal80/Cyo; C765-GAL4 ptc-lacZ/TM6B, Tb; ci^94^/Dp[y^+^]* were crossed to males of the genotype *yw hs-flp; FRT42D cos2^2^/Cyo; crCi-X/Dp[y^+^]*, selecting third instar larval progeny lacking *y^+^* and *Tb* to obtain wing discs with a single constructed *crCi* allele as the only source of Ci and GFP-marked clones lacking *cos2* activity.

Females of the genotype ('2R') *yw hs-flp UAS-GFP; smo^2^ FRT42D P[Smo^+^] tub-Gal80/Cyo; C765-GAL4 ptc-lacZ/TM6B, Tb; ci^94^/Dp[y^+^]* were crossed to males of the genotype *yw hs-flp; smo^2^ FRT42D UAS-GAP-Fu/Cyo; crCi-X/Dp[y^+^]*, selecting third instar larval progeny lacking *y^+^* and *Tb* to obtain wing discs with a single constructed *crCi* allele as the only source of Ci in GFP-marked clones expressing GAP-Fu and lacking *smo* activity.

Females of the genotype *yw hs-flp fu^mH63^; FRT42D P[y^+^] P[Fu+]/Cyo; (Su(fu)^LP^) C765-GAL4 ptc-lacZ/TM6B, Tb; ci^94^/Dp[y^+^]* were crossed to males of the genotype *yw hs-flp; Sp/Cyo; (Su(fu)^LP^/TM6B); crCi-X/Dp[y^+^]*, selecting male third instar larval progeny lacking *y^+^* and *Tb* to obtain wing discs lacking Fu kinase activity (with or without functional Su(fu)) and a single constructed *crCi* allele as the only source of Ci.

## Immunohistochemistry

Wing disc clones were generated by heat-shocking late first or early second instar larvae for 1 hr at 37°C and dissections took place 3.5 to 4 days later in wandering third instar larvae. Wing discs were dissected from late third instar larvae in PBS and fixed in 4% paraformaldehyde (in PBS) for 30 min, rinsed 3X with PBS, blocked with 10% normal goat serum (Jackson ImmunoResearch Laboratories, Inc) in PBS-T (0.1% Triton) for 1 hr, and stained with the following primary antibodies: rabbit anti–β-galactosidase (1:10,000; MP Biomedicals), mouse 4D9 anti-Engrailed (1:5 Developmental Studies Hybridoma Bank), Rat 2A1 anti-Ci (1:3 Developmental Studies Hybridoma Bank), overnight at 4°. Inverted Larvae were then washed three times in PBST for 10 min each and incubated with Alexa Fluor 488, 546, 594, or 647 secondary antibodies (1:1000; Molecular Probes) for 1 hr at room temperature. Larvae were washed twice in PBST for 20 min each, once in PBS for 10 min and mounted in Aqua/Poly mount (Polysciences).

## Quantitation from fluorescent images

Fluorescence images were captured using 20x, 63x, or 40x (discs with far anterior clones) objectives using 1.4 NA oil immersion lenses on a confocal microscope (LSM 700 and LSM800; Carl Zeiss). The range indicator was used to set the appropriate laser intensity per experiment for each fluorophore such that the signal was in the linear range.

Intensity Profiles: To measure intensity profiles along the AP axis, an elongated rectangle was drawn on a central region of the wing pouch, avoiding the D/V border. The y-axis shows the average fluorescence intensity over the height of the rectangle at each point on the x-axis (AP axis) for *ptc-lacZ* expression or Ci-155 protein, measured using Image J software (NIH, Bethesda, Maryland). In general, three wings discs per condition were measured and averaged for each plot, using the posterior edge of *ptc-lacZ* expression as a reference point for the AP border.

Clone Measurements: The average fluorescent intensity of *ptc-lacZ* or Ci-155 over specific regions was measured using Image J. Multiple clones or clone regions (for large clones), anterior regions (most commonly three per disc), AP border sections (three per disc), posterior regions (three per disc), were analyzed for each disc. To make sure the best region was acquired for measurements in clones, the region was selected using the GFP marker in the central part of the clone and confirmed to not be on a fold or shadowed region. For the AP border, regions were measured avoiding the DV boundary and abnormal folds. For *Figures 4* and *6*, *ptc-lacZ* clone intensity was calculated relative to AP border levels after subtracting anterior cell intensity values from each because *ptc-lacZ* is sometimes expressed artifactually in posterior cells: (clone-averaged anterior)/(averaged AP border-averaged anterior). In *Figure 5* *ptc-lacZ* intensity in clones and anterior cells outside clones was in each case divided by AP border intensity without any subtractions. Ci-155 clone intensity and intensity in anterior cells outside clones (*Figure 4*) were calculated relative to AP border levels after subtracting posterior cell intensity values from each: (clone-averaged posterior)/(averaged AP border-averaged posterior) and (anterior-averaged posterior)/(averaged AP border-averaged posterior). For *Figure 5*, *ptc-lacZ* intensity measurements in clones and anterior regions were divided by the AP-Border.

## Adult wings

Adult wings were pulled off anaesthetized flies and placed in 70% ethanol for 5 min, transferred to 100% ethanol, and then mounted in Aqua/Poly Mount (Polysciences). They were imaged with Transmitted Light on a Nikon Diaphot 300 microscope using a 10x objective.

## Statistics and reproducibility

All images shown are representative of at least five examples. No statistical method was used to predetermine sample size but we used prior experience to establish sufficient sample sizes. No samples were excluded from analysis, provided staining was of high quality. The experiments were not randomized; samples presented as groups in the results were often all part of the same experiment and were always treated in exactly analogous ways without regard to the identity of the sample. Investigators were not blinded during outcome assessment, but had no pre-conception of what the outcomes might be. For comparisons between the measured levels of *ptc-lacZ* product or Ci-155 a t-test was used to determine significance between pairs of genotypes (generally Ci variant versus wild-type Ci), and the errors for individual values determined from multiple samples was reported as the standard error of the mean.

## Acknowledgements

This work was supported by NIH RO1 GM041815 awarded to DK. We thank Aaron Choi, Jason Li and Sarah Finkelstein for research assistance, Hoyon Kim and other lab members for continued discussions and input, Rebecca Delker and Dr. Richard Mann for advice on CRISPR engineering, the Bloomington stock center for provision of genetic reagents, the Developmental Studies Hybridoma Bank (DSHB) for antibodies, FlyBase as an information resource, and the confocal microscope resource provided by the Dept. of Biological Sciences, Columbia University.

## Additional information

### Funding

| Funder | Grant reference number | Author |
|--------|------------------------|--------|
| National Institutes of Health | RO1 GM041815 | Daniel Kalderon |

The funders had no role in study design, data collection and interpretation, or the decision to submit the work for publication.

### Author contributions
Jamie C Little, Conceptualization, Resources, Data curation, Software, Formal analysis, Supervision, Validation, Investigation, Visualization, Methodology, Writing - original draft, Project administration, Writing - review and editing; Elisa Garcia-Garcia, Conceptualization, Resources, Software, Investigation, Methodology, Writing - review and editing; Amanda Sul, Investigation, Visualization, Writing - review and editing; Daniel Kalderon, Conceptualization, Resources, Data curation, Formal analysis, Supervision, Funding acquisition, Validation, Investigation, Methodology, Writing - original draft, Project administration, Writing - review and editing

### Author ORCIDs
Daniel Kalderon (iD) https://orcid.org/0000-0002-2149-0673

### Decision letter and Author response
Decision letter https://doi.org/10.7554/eLife.61083.sa1
Author response https://doi.org/10.7554/eLife.61083.sa2

## Additional files

### Supplementary files
• Transparent reporting form

### Data availability
All data reported in this study are included in the manuscript and supporting files.

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
