## [Decision Letter]

**Acceptance summary:**

Your manuscript addresses the regulation of the Ci transcriptional effector of the Hedgehog pathway: Ci is normally processed into a transcriptional repressor in the absence of ligand, but becomes an activator in the presence of Hedgehog. Your paper provides a novel view of the way the pathway functions: By showing that a Ci protein that cannot be processed still supports normal wing patterning shows that graded processing of Ci is not required: This indicates that there are two independent graded responses to the Hh signal and that regulation of full length Ci activity is sufficient to provide graded expression of target genes.

**Decision letter after peer review:**

Thank you for submitting your article "*Drosophila* Hedgehog can act as a morphogen in the absence of regulated Ci processing" for consideration by *eLife*. Your article has been reviewed by three peer reviewers, and the evaluation has been overseen by a Reviewing Editor and Utpal Banerjee as the Senior Editor. The following individual involved in review of your submission has agreed to reveal their identity: Bob Holmgren (Reviewer #2).

The reviewers have discussed the reviews with one another and the Reviewing Editor has drafted this decision to help you prepare a revised submission.

The reviewers are quite positive about the paper that makes the very strong point that it is possible to obtain graded activation of engrailed and patched with uniform repressor, supporting your model that graded processing of Ci is not required, and that regulation of full length Ci activity is sufficient to provide graded expression of target genes. This demonstrates that there are two independent graded responses to the Hh signal.

Yet, there are some points that need to be addressed before the paper can be published:

– Ci degradation: Do you know that this is not due to transcriptional regulation of Ci itself? If you have not tested this, you should be more cautious in the way you present these data : If it is Ci transcription that is changing, you might be measuring regions with very high Hh signaling, or looking at constructs that do not support high signaling. You should test whether a *UAS-Ci* construct has weaker Ci in the zone of supposed degradation.

– Your work yields new questions about how Cos2 and PKA regulate Ci.

Attributing the effects of deleting CORD in PKA mutants to the lack of Cos2 binding is speculative (this is even in the Abstract). Although this is a potential explanation, other factors may be at work and must be discussed. You should look at whether *cos2* clones do or do not increase signaling in the background carrying the processing-deficient δ-CORD to show that the CORD domains are critical for Cos2-based inhibition. So far, we only know that removing the CORD domain strengthens the unknown PKA effect independently from Cos2.

– The results from Seoung showed that the boundary reduction of Ci is unaffected in clones homozygous for a Hib nonsense mutant. So it is not surprising that a mutant Ci that does not bind Hib is still reduced in the boundary zone. You must acknowledge and discuss explicitly any discrepancy with the Seoung et al. results.

– In general, the manuscript is very dense and difficult to follow, and extensive text editing will be necessary to clarify and address many of the reviewers' comments.

Reviewer #1:

The manuscript extends previous work by the author and others on the activity of Hh pathway in *Drosophila* wing discs, and how Hh components affect the activity and levels of the Ci transcription factor. While most previous work has been performed using altered UAS-Ci constructs, the present study has instead used engineered constructs expressed from endogenous Ci promoters; these express at endogenous levels, avoiding some possible artifacts of previous studies. They use mutant Ci constructs first to show that spatial regulation of Ci processing is not necessary for normal patterning. They then exploit processing deficient and other forms of Ci to examine processing-independent activities of Fu, Su(fu) and Cos2. The manuscript is dense and occasionally speculative, especially in these latter experiments. But the major points are certainly of interest to general readers. However, I had substantial difficulty with several portions of the manuscript that I think need to be clarified before I would recommend acceptance.

1) The authors spend some time examining a region of the disc with low Ci-155 levels that is induced by very high Hh levels and that most authors assume is caused by wholesale degradation of Ci, rather than processing into its repressor form. Here they show that Ci constructs that cannot bind to Hib, a Hh-induced ubiquitin ligase that is thought to degrade Ci, still have reduced levels just anterior to the AP where Hh signaling is high. However, a previous study has already shown that Ci levels remain low in this region after genetic removal of Hib (Seoung et al., 2010 Figure 3) so this result is not unexpected.

2) What is the evidence that this region of low Ci is caused by degradation? I ask because Blair '82 (Development) showed that at pupal stages the transcriptional marker ci-plac is reduced in this region; given the stability of βGal this could indicate an earlier loss of Ci transcription. Is there a published in situ of sufficient detail to resolve this point? Alternatively, is the region of reduced Ci seen using *UAS-Ci* constructs that are not regulated by Ci enhancers?

3) I had difficulty following the logic in the CORD domain section of the manuscript. Although the reasoning is not explained, the conclusion seems to be based on the idea that any Ci construct that increases the effects of a PKA mutant clone must have done so by removing the inhibitory activity of Cos2. Since Ci without the CORD domain has a stronger effect in PKA clones, the authors conclude that this form of Ci is not inhibited by Cos, and thus CORD is critical for Cos2 interactions. But is this really a valid assumption? Might not the CORD domain act in some other way?

The constructs that lack Cos2 binding domains can also be processed, so unlike the previous section the authors are not examining PKA and Cos2 effects that are independent of processing. Yet the Discussion seems to assume that this effect is the same as the processing-independent effect. While the authors do show that a Ci that lacks both processing and the CORD domain shows increased signaling, they do not examine whether this is mediated by PKA or Cos2.

Reviewer #2:

In Little et al. the authors generated a series of ci constructs, which are expressed at near physiological levels using either the endogenous promoter region or ci promoter transgenes. They use these constructs to assay various aspects of Hh signal transduction. There are two important conclusions in this paper; the first of which is profound. By combining a ci gene that encodes a protein that can't be processed with the ci^Ce^ mutation, which constitutively expresses a repressor like form of Ci, normal wing patterning is restored. Thus, graded processing of Ci into the repressor is not required, and regulation of full-length Ci activity is sufficient to provide graded expression of target genes. This demonstrates that there are two independent graded responses to the Hh signal. The second result that Hib mediated proteolysis of Ci does not appear to be responsible for decreased Ci levels along the compartment boundary is also quite interesting and surprising. These data are excellent, and the experiments are well controlled, so the work does warrant publication in *eLife*.

The remaining experiments are less novel. GAP-Fu was shown to activate *ptc-lacZ* in Claret et al., 2007. In this paper the authors show that this effect responds to Ci protein levels.

The experiments examining regulation by PKA and Cos2 are worthwhile as this lab has shown in the past that overexpression experiments looking at Hh pathway signaling can be subject to artifacts. The authors confirm that PKA and Cos2 can negatively regulate Ci away from the compartment boundary and show that this regulation is not dependent on the three PKA sites involved in proteolytic processing (for PKA regulation) or the CDN and CORD domains for Cos2 regulation. A curious result is that deletion of the CORD domain augmented the activation of *ptc-lacZ* in PKA clones.

Reviewer #3:

This manuscript focuses on regulation of the Hedgehog (Hh) transcriptional effector Ci, which is processed to a truncated transcriptional repressor in the absence of ligand, and to a potent transcriptional activator in its presence. Studies are aimed at understanding how the Hh morphogen gradient is interpreted through Ci protein regulation in the *Drosophila* wing imaginal disc. Although this is a research area that has received much attention, prior studies were often performed using over-expressed proteins, calling into question whether the observed regulatory processes were indicative of how the protein is regulated at physiological levels. The strength of this manuscript is the use of CRISPR to alter the endogenous Ci gene so that studies could be carried out on physiologically-expressed protein in an in vivo system. Studies reveal that the Hh morphogen gradient can control activity of Ci to ensure proper wing development, even when regulated processing does not occur. The authors report that in addition to promoting processing of Ci to its truncated repressor species, Costal2 and PKA also contribute to Hh gradient interpretation by controlling Ci-155 activator function.

Overall, this is a well-executed study that corrects previous assumptions made about how the Ci activity gradient is controlled. The only experimental request I have is that a little more investigation be performed to determine how PKA is inhibiting Ci-155 activity if not through phosphorylating it to control repressor formation. A Ci PKA-insensitive mutant that lacks the phosphorylation sites to facilitate its conversion to repressor, can be further activated following PKA loss. There is speculation that an additional two known PKA sites might contribute to control of activity, but a Ci CORD domain mutant that lacks those 2 additional sites can also be activated by PKA loss. Can you determine whether this additional repression function by PKA depends upon its kinase activity, or is it a scaffolding/binding activity? Can you investigate what happens with a kinase dead PKA? Perhaps in clones?

---

## [Author Response]

The reviewers are quite positive about the paper that makes the very strong point that it is possible to obtain graded activation of engrailed and patched with uniform repressor, supporting your model that graded processing of Ci is not required, and that regulation of full length Ci activity is sufficient to provide graded expression of target genes. This demonstrates that there are two independent graded responses to the Hh signal.Yet, there are some points that need to be addressed before the paper can be published:– Ci degradation: Do you know that this is not due to transcriptional regulation of Ci itself? If you have not tested this, you should be more cautious in the way you present these data : If it is Ci transcription that is changing, you might be measuring regions with very high Hh signaling, or looking at constructs that do not support high signaling. You should test whether a UAS-Ci construct has weaker Ci in the zone of supposed degradation.

We have changed the description of reduced Ci-155 levels close to the source of Hh (or reductions in other high pathway situations, such as GAP-Fu clones) to the literal observation of “Ci-155 reduction”. As reviewer 1 states, the reduction has commonly been attributed to full proteolytic degradation. That is a reasonable portrayal if the loss of Ci-155 is due to Rdx/Hib/Cul3 binding to Ci-155, followed by ubiquitination and proteolysis. However, that has never been demonstrated convincingly and the results we present suggest that is not the major means of Ci-155 reduction. Our evidence suggests that Su(fu) is necessary to observe reduced Ci-155 in this region. We suspect that is because Su(fu) protects Ci-155 from degradation and Hh reduces Ci/Su(fu) interactions but that remains a hypothesis. We therefore agree with the reviewer’s point that it is not appropriate to assume that the reduced Ci-155 is due to degradation, although that remains the most likely explanation. We discuss these issues and past findings relevant to potential non-proteolytic mechanisms in the Discussion.

The relevant results that we report are (i) the profile of CI-155 reduction, (ii) the lack of response to altering Rdx/Hib binding sites and (iii) the consequences of removing Su(fu). None of these conclusions requires knowledge or further exploration of the mechanism(s) underlying Hh-stimulated Ci-155 reduction.

The suggested experiment of using *UAS-Ci* to remove the variable of ci transcription is unfortunately not feasible because *UAS-Ci* transgenes do not support normal pathway activity (Garcia et al., 2017). That is the reason we are using CRISPR ci alleles (and the large amount of extra work that entails) to study the activity of Ci variants.

– Your work yields new questions about how Cos2 and PKA regulate Ci.Attributing the effects of deleting CORD in PKA mutants to the lack of Cos2 binding is speculative (this is even in the Abstract). Although this is a potential explanation, other factors may be at work and must be discussed. You should look at whether cos2 clones do or do not increase signaling in the background carrying the processing-deficient δ-CORD to show that the CORD domains are critical for Cos2-based inhibition. So far, we only know that removing the CORD domain strengthens the unknown PKA effect independently from Cos2.

We have added to the discussion of inferences from deletion of the CORD domain along the lines suggested (including adding “likely” to the Abstract, internal headings and conclusions). We present the deduction that Cos2 inhibits Ci-155 by binding the CORD domain as the simplest and most likely explanation of our results, both at first mention and later.

I believe a key issue, not mentioned in the reviewers’ summary above, is that loss of the CORD domain increases the activity of processing-resistant Ci (with normal PKA and Cos2 activity). Loss of Cos2 also increases the activity of processing-resistant Ci. Loss of the CORD domain does not, however, change activity in cos2 mutant clones where there is no processing and no Cos2-CORD binding. The simplest explanation is that loss of cos2 and loss of CORD affect the same inhibitory interaction, namely Cos2-CORD binding. This line of reasoning and data always compare the activities of processing-resistant Ci and are all with normal PKA activity.

Testing processing-deficient Ci lacking the CORD domain in cos2 mutant clones would not test anything new (we already tested the effect of loss of CORD in cos2 mutant clones and there is no processing of any type of Ci in cos2 mutant clones). We did, nevertheless, intend to make the suggested test in order to be thorough. However, the double mutant Ci variant, as with other hyperactive variants, could not be manipulated to yield suitable animals to make the test despite several attempts.

The observation that Ci lacking the CORD domain had higher activity than wild-type Ci in pka mutant clones provides further evidence of the CORD domain being inhibitory when Ci-155 is not processed (this time in a situation where there is no PKA activity). This evidence is discussed first in the Results but the increased activity of processing-resistant Ci when CORD is deleted is more straightforward (because it does not involve the unknown consequences of eliminating PKA).

– The results from Seoung showed that the boundary reduction of Ci is unaffected in clones homozygous for a Hib nonsense mutant. So it is not surprising that a mutant Ci that does not bind Hib is still reduced in the boundary zone. You must acknowledge and discuss explicitly any discrepancy with the Seoung et al. results.

We now clarify that our test is different from all prior tests because it examines the consequences of direct actions of Rdx/Hib on Ci (by removing Rdx/Hib binding sites), rather than all actions of Rdx/Hib (by eliminating Rdx/Hib). The result (no significant change of Ci-155 in the posterior half of the AP border) was similar to the result of Seong examining all Rdx/Hib actions in null clones (which we cited previously). We also cite the two other studies that presented different results for Rdx/Hib clones. The discrepancy among Rdx/Hib clone phenotypes is present in the literature (the images themselves are not necessarily all compelling but the written conclusions are clearly different) and we do not attempt to resolve it. Our result, concerning the direct effects of Rdx/Hib on Ci is potentially consistent with either result concerning the net effect of Rdx/Hib because Rdx/Hib potentially can also reduce Ci-155 levels indirectly. There is certainly no discrepancy with the Seong result to discuss. We present our results literally, with appropriate caveats, as a contribution towards eventual resolution of what exactly Rdx/Hib accomplishes. We do not make a major conclusion on this, larger issue.

– In general, the manuscript is very dense and difficult to follow, and extensive text editing will be necessary to clarify and address many of the reviewers' comments.

We have re-written many sections, guided by specific reviewer suggestions and acknowledged difficulties, trying to separate relevant prior evidence and steps of complicated arguments into single sentences and in a logical order. The net result is that several dense sentences have been expanded into simpler sets of sentences and we have tried at each step to be clear about distinctions between evidence and assumptions.

Reviewer #1:The manuscript extends previous work by the author and others on the activity of Hh pathway in *Drosophila* wing discs, and how Hh components affect the activity and levels of the Ci transcription factor. While most previous work has been performed using altered UAS-Ci constructs, the present study has instead used engineered constructs expressed from endogenous Ci promoters; these express at endogenous levels, avoiding some possible artifacts of previous studies. They use mutant Ci constructs first to show that spatial regulation of Ci processing is not necessary for normal patterning. They then exploit processing deficient and other forms of Ci to examine processing-independent activities of Fu, Su(fu) and Cos2. The manuscript is dense and occasionally speculative, especially in these latter experiments. But the major points are certainly of interest to general readers. However, I had substantial difficulty with several portions of the manuscript that I think need to be clarified before I would recommend acceptance.1) The authors spend some time examining a region of the disc with low Ci-155 levels that is induced by very high Hh levels and that most authors assume is caused by wholesale degradation of Ci, rather than processing into its repressor form. Here they show that Ci constructs that cannot bind to Hib, a Hh-induced ubiquitin ligase that is thought to degrade Ci, still have reduced levels just anterior to the AP where Hh signaling is high. However, a previous study has already shown that Ci levels remain low in this region after genetic removal of Hib (Seoung et al., 2010 Figure 3) so this result is not unexpected.

This is addressed above. I imagine that many individuals with an interest in this subject might express a belief, based on earlier papers, that Rdx/Hib is responsible for reducing Ci-155 in posterior regions of the AP border despite the results of Seong (and perhaps also after reading our current report). It is therefore certainly worth highlighting our result and past discrepancies.

2) What is the evidence that this region of low Ci is caused by degradation? I ask because Blair '82 (Development) showed that at pupal stages the transcriptional marker ci-plac is reduced in this region; given the stability of βGal this could indicate an earlier loss of Ci transcription. Is there a published in situ of sufficient detail to resolve this point? Alternatively, is the region of reduced Ci seen using UAS-Ci constructs that are not regulated by Ci enhancers?

In addition to what was addressed above, we have included reference to prior work on RNA in situs and ci-lacZ expression and discuss the possibility of ci regulation at different levels. It seems plausible that En induced at late larval stages might reduce ci transcription (to give the quoted pupal ci-lacZ pattern). Older RNA in situ (and ci-lacZ) for 3^rd^ instar (and our own unpublished results for ci-lacZ) suggest uniform transcription but I am not aware of RNA in situs of high resolution with really good quantitation. We may examine that in the future but the main response to the issue raised is to acknowledge that the mechanism(s) for Ci-155 reduction are not clear and may not all involve Ci degradation. Our revised discussion includes the issues and observations raised by the reviewer, which altogether served as an important check on the state of evidence and assumptions underlying the observed reduction of Ci-155 under conditions of high Hh pathway activity.

3) I had difficulty following the logic in the CORD domain section of the manuscript. Although the reasoning is not explained, the conclusion seems to be based on the idea that any Ci construct that increases the effects of a PKA mutant clone must have done so by removing the inhibitory activity of Cos2. Since Ci without the CORD domain has a stronger effect in PKA clones, the authors conclude that this form of Ci is not inhibited by Cos, and thus CORD is critical for Cos2 interactions. But is this really a valid assumption? Might not the CORD domain act in some other way?The constructs that lack Cos2 binding domains can also be processed, so unlike the previous section the authors are not examining PKA and Cos2 effects that are independent of processing. Yet the Discussion seems to assume that this effect is the same as the processing-independent effect. While the authors do show that a Ci that lacks both processing and the CORD domain shows increased signaling, they do not examine whether this is mediated by PKA or Cos2.

This has been addressed above. I believe that the caveats the reviewer mentions are acknowledged and that the conclusion we present as the most likely explanation is indeed the most likely. I cannot think of a strong competitor and think that any further speculations would serve only to confuse readers. I think it is unlikely that Cos2 inhibits Ci-155 through a mechanism other than direct binding. Removing the CDN domain has no effect, while removing the CORD domain has an effect equivalent to removing Cos2 (when there is no processing) and does not increase activity further in a cos2 mutant clone (where there is no processing). It would be nice to test the zinc finger region, but we never found an alteration that eliminates Cos2 binding without affecting DNA binding.